# Natural hazard fatalities in Switzerland from 1946 to 2015

Alexandre Badoux[1], Norina Andres[1], Frank Techel[2], Christoph Hegg[1]

[1]Swiss Federal Research Institute WSL, Zürcherstrasse 111, CH-8903 Birmensdorf, Switzerland
[2]WSL Institute for Snow and Avalanche Research SLF, Flüelastrasse 11, CH-7260 Davos, Switzerland

*Correspondence to*: Alexandre Badoux (badoux@wsl.ch)

**Abstract.** A database of fatalities caused by natural hazard processes in Switzerland was compiled for the period between 1946 and 2015. Using information from the Swiss flood and landslide database and the Swiss destructive avalanche database, the data set was extended back in time and more hazard processes were added by conducting an in-depth search of newspaper reports. The new database now covers all natural hazards common in Switzerland categorized into seven process types: flood, landslide, rockfall, lightning, windstorm, avalanche, and other processes (e.g. ice avalanches, earthquakes). Included were all fatal accidents associated with natural hazard processes where victims did not expose themselves to an important danger on purpose or wilfully. The database contains information on 635 natural hazard events causing 1023 fatalities, which corresponds to a mean of 14.6 victims per year. The most common causes of death were snow avalanche (37%), followed by lightning (16%), flood (12%), windstorm (10%), rockfall (8%), landslide (7%) and other processes (9%). About 50% of all victims died in one of the 507 single-fatality events; the other half of victims were killed in the 128 multi-fatality events.

The number of natural hazard fatalities that occurred annually during our 70-year study period ranged from two to 112 and exhibited a distinct decrease over time. While the number of victims during the first three decades (until 1975) ranged from 191 to 269 per decade, it ranged from 47 to 109 in the four following decades. This overall decrease was mainly driven by a considerable decline in the number of avalanche and lightning fatalities. About 75% of victims were males in all natural hazard events considered together, and this ratio was roughly maintained in all individual process categories except landslides (lower) and other processes (higher). The ratio of male to female victims was most likely to be balanced when deaths occurred at home (in or near a building), a situation that mainly occurred in association with landslides and avalanches. The average age of victims of natural hazards was 35.9 years, and accordingly, the age groups with the largest number of victims were the 20-29 and 30-39 year-old groups, which in combination represented 34% of all fatalities. It appears that the natural hazard fatality rate in Switzerland during the past 70 years has been relatively low in comparison to rates in other countries or rates of other types of fatal accidents in Switzerland.

Keywords: natural hazard fatality, fatality rate, flood, landslide, avalanche, loss of life, natural disaster

## 1. Introduction

Every year, world-wide natural hazard events not only generate tremendous financial damage costs but also cause a large number of human fatalities (MunichRe, 2015). According to the NatCatSERVICE database of MunichRe, the average annual global loss of life due to natural catastrophes was 68,000 during the last ten years and 54,000 during the last thirty years (23,000 in 2015; Ins. Inf. Inst., www.iii.org/fact-statistic/catastrophes-global).

In scientific literature, information and data sets on loss of human lives due to specific natural hazard processes cover various time periods and exist at different aggregation levels: at the global scale (e.g. Jonkman, 2005; Petley, 2012; Auker et al., 2013; Dowling and Santi, 2014), the continental scale (e.g. Di Baldassarre et al., 2010; Sepúlveda and Petley, 2015) and most commonly the regional/national scale (e.g. Guzzetti, 2000; Ashley and Ashley, 2008; Vranes and Pielke, 2009; Singh and Singh, 2015). Moreover, there are studies that describe the circumstances during specific catastrophic natural hazard events and/or assess the patterns and reasons behind the associated massive loss of life (e.g. Chowdhury et al., 1993; Tsai et al., 2001; Doocy et al., 2007; Jonkman et al., 2009; Ando et al., 2013).

While some authors have analysed natural hazard mortality data that include many hazard types (e.g. Shah, 1983; Noji, 1991; Borden and Cutter, 2008), the bulk of studies have focussed on a distinct hazard process. Jonkman (2005) studied statistics about loss of human life caused by various freshwater flood types (river floods, flash floods and drainage problems) on a global scale and for the period from 1975 to June 2002 based on the EM-DAT International Disaster Database. This investigation showed that while flash flood events have the highest average fatality rate (deaths divided by number of affected persons), Asian river floods are most devastating in terms of the number of persons killed or affected. In addition, very high death tolls have been reported for coastal flood events (e.g. Chowdhury et al., 1993) that were not included in Jonkman (2005). On a national scale, flood fatalities have been assessed by many authors for countries all around the world, such as the USA (Ashley and Ashley, 2008), India (Singh and Kumar, 2013), Pakistan (Paulikas and Rahman, 2015) and Australia (Coates, 1999; FitzGerald et al., 2010).

Petley (2012) assembled a global data set of fatalities from non-seismically triggered landslides that took place from 2004 to 2010 based on the Durham Fatal Landslide Database (DFLD). The total number of landslides and fatalities during the 7-year period turned out to be an order of magnitude larger than numbers suggested by other sources, and the study indicated that most fatalities occur in Asia. Most recently, Sepúlveda and Petley (2015) published a new data set of landslides that caused loss of life in Central and South America as well as in the Caribbean. This continental analysis used an enhanced version of the DFLD and applied key search terms in Spanish. In a study focusing on fatalities caused by debris

flows (often included in landslides studies), Dowling and Santi (2014) considered 213 events that occurred between 1950 and 2011 and during which a total of 77,779 people were killed. Results of this global analysis provided evidence that more debris-flow fatalities tend to occur in developing countries. On a national scale, landslide events with fatal consequences were compiled by Guzzetti (2000) for events that occurred in contemporary Italy from 1279 to 1999. Fast mass movements, such as rockfall events, rockslides, rock avalanches and debris flows were included in this study and were found to have caused the largest number of fatalities.

Loss of life related to meteorological hazard events, such as lightning and all the different types of wind storms, has also been the subject of many national and regional studies. In the USA, medium- to long-term data sets of wind related deaths have been investigated, for example for tornados (Ashley, 2007) and hurricanes (Rappaport, 2000; Czajkowski et al., 2011), as well as for (nontornadic) convective (Black and Ashley, 2010) and nonconvective (Ashley and Black, 2008) high-wind events. Recent publications have presented national lightning data sets, for example from the UK (Elsom, 2001), India (Singh and Singh, 2015), Australia (Coates et al., 1993), Colombia (Navarrete-Aldana et al., 2014), USA (López and Holle, 1996; Curran et al., 2000), and Swaziland (Dlamini, 2009).

Geophysical events such as tsunamis and volcanic activity might not be very relevant for a country like our study area Switzerland, but are of considerable importance when assessing consequences of natural hazards at a global level (e.g. Auker et al., 2013). Earthquakes, in contrast, have occurred in Switzerland with dramatic effects (Fäh et al., 2009), but events causing fatalities or large amounts of damage are rare. For countries more frequently struck by seismic activity, various fatality databases exist (e.g. Vranes and Pielke, 2009).

In Switzerland, fatalities caused by floods, debris flows, and landslides have been systematically collected since 1972. They are recorded in the Swiss flood and landslide damage database (Hilker et al., 2009) and were briefly analysed by Schmid et al. (2004). Deaths due to rockfall incidents have been included in the database since 2002. For disastrous events causing loss of life that occurred before 1972, only partial and scattered information is available (e.g. Röthlisberger, 1991). In parallel, information on loss of life caused by snow avalanches in Switzerland has been collected since the hydrological year 1936/1937 by the WSL Institute for Snow and Avalanche Research SLF. To our knowledge, no systematic data acquisition of fatalities and damage induced by convective and non-convective high winds and lightening has been carried out at national scale in Switzerland.

In the study presented here, we compiled the available data on natural hazard fatalities mentioned above, extended the period covered by the database, and expanded it to include all process types relevant for the study area. We present this new, detailed 70-year (1946-2015) data set of loss of life in Switzerland caused

by (i) floods, (ii) debris flows, (iii) landslides and hillslope debris-flows, (iv) rockfall events and rockslides, (v) windstorms, (vi) lightning strikes, (vii) earthquakes and (viii) avalanches. Temporal and spatial patterns in the results are discussed and the numbers of fatalities as well as the characteristics of the underlying incidents for different process types are assessed. Finally, we compare our data with similar data from other countries and regions and attempt to explain differences qualitatively.

According to the Federal Constitution of the Swiss Confederation, the cantons and municipalities are responsible to ensure the protection of the population against natural hazards. The aim of this study is to support authorities to better understand the occurrence of fatal incidents, to identify potential improvements in hazard prevention and to further reduce the number of victims of natural hazards.

## 2. Study area

Switzerland is located in central Europe between latitudes 45° and 48° N, and between longitudes 5° and 11° E, with a total area of 41,285 km$^2$ and an altitudinal range of 193 to 4634 m a.s.l. The Swiss Confederation consists of 20 cantons and 6 half cantons, and its territory can be roughly divided into four regions based on its geomorphology (see Figure 1): the Alps (a high-altitude mountain range running across the central-south of the country), the Swiss Plateau (a relatively flat area between Lake Geneva and Lake Constance), the Prealps (the transitional area between the Alps and the Swiss Plateau) and the Jura (a hilly mountain range in the north-west). The population grew from 4.5 to 8.3 million people between 1946 and 2015 and is clustered mostly on the Swiss Plateau (over 60% of the total population is located on less than a third of the total area of the country). The Swiss climate is temperate but can vary regionally. In large parts of the Alps the mean annual rainfall is around 2000 mm/year or more, and along the Swiss Plateau this value amounts to 1000-1500 mm/year. Most precipitation falls during the summer months.

## 3. Data and methods

The data for this study were extracted from the Swiss flood and landslide damage database (section 3.1) and the Swiss destructive avalanche database (section 3.2). The data set (except for avalanches) was then extended (i) in time to include an additional period of 26 years back to the year 1946 and (ii) in breadth to include additional relevant natural hazard processes, such as lightning, windstorm, and earthquake, by carrying out an extensive newspaper search (sections 3.3 and 3.4).

### 3.1 Swiss flood and landslide damage database

Since 1972, fatalities and estimates of financial damage costs caused by naturally triggered floods, debris flows, landslides and (since 2002) rockfall events have been collected by the Swiss Federal Research Institute WSL in the Swiss flood and landslide damage database (Hilker et al., 2009; Badoux et al., 2014). Fatality and damage information is primarily provided by approximately 3000 Swiss newspapers and

magazines, which are scanned daily by a media-monitoring company. Additional information is often compiled from insurance companies and the websites of public authorities such as police and fire brigades. An in-depth description of the structure of the Swiss flood and landslide damage database was presented by Hilker et al. (2009). For this study, the database provided information on a total of 129 deaths due to floods, debris flows, landslides, hillslope debris-flows or rockfall from 1972 to 2015. In a next step, this basic data set was expanded using data on avalanche victims.

### 3.2 The Swiss destructive avalanche database

Data on avalanche casualties since the winter 1936/37 are stored in the destructive avalanche database from the WSL Institute for Snow and Avalanche Research SLF (Techel et al., 2015). In the 79 years from 1936/37 to 2014/15, 1255 avalanches killed 1961 people in Switzerland. We retrieved data for our study period from this database and considered cases that occurred (i) in settlement areas, (ii) on high-alpine building sites, (iii) on transportation corridors (including roads and railway lines, ski runs and winter hiking trails, if any of these were officially open or if they were closed but the casualty was work related), and (iv) on hiking trails during summer if the trail was open and snow-free.

We explicitly excluded all cases that occurred outside of transportation corridors and settlements (except hiking trails in summer, see (iv) above). Thus, cases associated with ski or snowshoe touring, and skiing or snowboarding away from open ski trails were not included in our data set. Also, we did not incorporate avalanche fatalities related to vehicles illegally driving on officially closed roads. The destructive avalanche database is considered complete for fatality data, and data quality is generally very high. Thus, no systematic search of avalanche related events in a newspaper was necessary.

### 3.3 Further extension of the record of natural hazard fatalities in Switzerland

With the aim of extending the data series to 70 years and combining it with other natural hazard processes like windstorm, lightning, earthquake and ice avalanche, we made a search in a newspaper. We selected the „Neue Zürcher Zeitung“ (NZZ) because it is a national newspaper and a digital archive exists back until 1780. The NZZ is written in German, which is the most spoken language in Switzerland (see also section 5.1). We  accessed the digital archive via an internet platform, where a keyword search was possible. In a first step, we derived adequate keywords (in German) for the search. We selected the years 1986-1995, for which we already had some data from the Swiss flood and landslide database, as validation period. We generated a list with possible keywords and checked how often these words were used in the newspaper for the description of casualties that occurred in Switzerland and abroad. The casualties from abroad were included to get more search hits. We then shortened the list to the most relevant keywords. Where possible, we combined the keywords for the different processes. For example, we searched for casualties caused by “flood OR inundation OR landslide OR landslip OR mudslide OR mudflow” combined with “dead OR casualty OR death OR human life OR drown OR killed OR dead body OR buried”. An overview of the

keywords and the combinations used is given in the supplementary material. With these keyword combinations, we found most cases (roughly 90%) already stored in the Swiss flood and landslide database for the selected validation period 1985-1995.

In a next step, we used the selected keywords to scan the newspaper for the remaining years (1946-1977, 1988-2015). For the years where we already had data from the Swiss flood and landslide database for the processes flood, landslide, debris flow and rockfall, we restricted the search to the processes not already covered. The search for fatalities produced up to 300 hits per year. We initially viewed all hits, but many were not relevant for our research (e.g. fatalities abroad). Further, the digital scan of the newspaper was sometimes of bad quality, which resulted in the misspelling of words and thus influenced our search because not all of our keywords were found. In addition, some gaps exist in the digital archive of the NZZ (e.g. 04. – 15.08.1978).

For each casualty found, a database entry was made. The following information describing the fatal accident and the victim was stored: name of municipality, canton, date, time, coordinates, description of the event, age, gender, locality (i.e. in or around a building, on a transportation route, in open terrain, in a stream channel, on a lake), mode of transport (on foot, by bicycle, in vehicle, by public transport, by boat, by ski), activity (work, leisure time), and data source. For most of the above-mentioned categories, the quality of the information was also assessed. In doing so, we distinguished between two types of information quality: (i) a concise statement describing a certain characteristic of the event (certain); and (ii) an indication of a characteristic that we deduced based on the available information (probable). In contrast, if no information was available to describe certain aspects of a fatal incident, those characteristics were considered unknown.

### 3.4 Natural hazard processes considered in the new database

In the present study, we assigned the fatalities found in our search (or adopted from the flood and landslide damage database or the destructive avalanche database) to the following seven process categories:

- Flood: includes people drowned in flooded or inundated areas or carried away in streams under high-water conditions.
- Landslide: includes people killed by landslides, hillslope debris flows and channelized debris flows. Because debris flows were often not identified as such in the press media (especially in the first half of our study period), we decided to add debris flow fatalities to the category of landslide processes. This approach has been applied previously, for example by Guzzetti (2000).
- Rockfall: includes people killed by rockfall.
- Lightning: includes all people who died after being struck by lightning.

- Windstorm: includes people killed by falling objects or trees during very strong wind conditions and people who drowned in lakes because their boat capsized during such conditions.
- Avalanche: includes people killed in snow avalanches (except roof avalanches, see below).
- Other: includes all people killed by hazard processes that are not frequent in Switzerland (e.g. ice avalanches, earthquakes, lacustrine tsunamis, roof avalanches). Most of the fatalities assigned to this process type were caused by the 1965 Mattmark ice avalanche.

Fatalities due to forest fires did not occur during our study period, and people who died during meteorological heat waves were not included. Overall, we considered only casualties where people did not expose themselves to a considerable danger on purpose or wilfully. For example, we excluded loss of life due to high-risk sports (e.g. canoeing and river surfing during floods) and other outdoor activities in potentially dangerous environments, such as canyoning, mountaineering and rock climbing. We also excluded popular snow sports experienced outside of ski resorts, such as freeriding and alpine touring, that have been described elsewhere (e.g. Techel and Zweifel, 2013; Schweizer and Lütschg, 2001). Further, we only included cases where the process directly induced a casualty or an action that led to death. For example, we did not consider cases where a forest ranger was killed during forest clearing operations after a windstorm or where a firefighter was killed in a flooded basement due to an electric shock.

## 4. Results

### 4.1 Types of natural hazard processes associated with fatalities

Our newly compiled database includes reports on 1023 fatalities associated with natural hazard processes in Switzerland during the period from 1946 to 2015 (Table 1). This result corresponds to an average of 14.6 fatalities per year. More than one third of all fatalities (378 deaths) were caused by snow avalanches during winter and spring. The second most frequent cause of loss of life was lightning (16.0%), followed by flood (12.1%) and windstorm (10.3%). Landslides and rockfall events each represented less than 10% of the total number of fatalities in Switzerland. Processes that caused sporadic deaths included an earthquake (3 deaths), a lacustrine tsunami (1) and a roof avalanche (1). The worst incident, in terms of the number of fatalities involved, that occurred during our 70-year study period was the catastrophic 30 August 1965 ice avalanche, which broke off at the terminus of the Allalin Glacier in the canton of Valais, destroyed the Mattmark Dam construction site and killed 88 people. This incident was the only deadly ice avalanche event we considered in the database (and is included in the category *other* processes, Table 1).

### 4.2 Temporal distribution of natural hazard fatalities in Switzerland

#### 4.2.1 Annual distribution of fatalities

The annual number of natural hazard fatalities in Switzerland during our 70-year study period ranged from two (in five years and most recently in 2010) to 112 (in 1951; Figure 2). The resulting median over the

entire period was 9.0 deaths per year, which is below the mean value of 14.6 and highlights the large influence of severe multi-fatality events (see also section 5.2). While two years had an annual number of deaths greater than 100, a total of five years exceeded the value of 33.2 (mean plus one standard deviation). The annual number of events ranged from one (in 1995) to 45 (in 1951), with a median over the study period of 7.0 and a mean value of 9.1.

The number of people killed by natural hazards in the last 70 years showed a clear decrease over time (Figure 2). The downward trend in the total number of annual fatalities is statistically significant (Mann-Kendal test: Tau = -0.42, 2-sided P-value = 3.35e-7; Linear Regression Model: 2-sided P-value = 0.001). This pattern is confirmed when the total number of deaths during the first 35 years of the study period (747) is compared with the value for the second 35 years (276). Thus, nearly three times as many people were killed by natural hazard processes from 1946 to 1980 than from 1981 to 2015. Further, only three years after 1981 exhibited a number of fatalities larger than the mean value for the full 70-year period: 1985, 1999 and 2000. The decrease in natural hazard fatalities over the 70-year period is also apparent in the number of fatalities per decade (Figure 3). On average, nearly three times as many people died in accidents during each of the first three decades of the study period as during each of the last four decades.

The temporal distributions of victims of specific hazard types showed a distinct decrease for lightning and avalanches only (Figures 3 and 4). The trend is statistically significant for avalanches (Mann-Kendal test: Tau = -0.30, 2-sided p-value = 0.005; Linear Regression Model: 2-sided p-value = 0.0) and lightning (Mann-Kendal test: Tau = -0.28, 2-sided p-value = 0.008; Linear Regression Model: 2-sided p-value = 0.014). For both process types, around four times more fatalities were recorded in the first half of the study period than in the second half. Interestingly, during the last 15 years, the database revealed only seven avalanche fatalities, resulting in an average for this most recent period that is ten times smaller than the overall mean value. In a qualitative sense, only deaths due to landslide processes seem to have increased slightly over the 70-year period. However, this impression is was strongly influenced by the large number of fatalities (16) that occurred during the severe October 2000 event in the canton of Valais, when 13 people died in the Gondo landslide.

There were several years in the data set of each natural hazard type when no fatal incidents were reported (Figure 4). The three hazard types for which at least one fatality was reported in most years during the study period are lightning (49 years), avalanches (48 years) and floods (48 years). While at least one fatality associated with windstorm and rockfall events occurred in 40 and 37 years, respectively, fatal accidents related to landslide processes occurred only in one third of the investigated years. Even though only one deadly ice avalanche happened between 1946 and 2015, this event was responsible for more fatalities than all rockfall or all landslide incidents.

Normalization of fatality data by population resulted in a clearly declining annual fatality rate (Figure 5). We found an annual average rate of 3.9 deaths per million persons for the first 35 years of the study period and a rate of 1.1 for the second 35 years. The yearly mean for the whole period is 2.5 victims per million persons. A very distinct decrease in the fatality rate from the first to the second half of the study period can be seen for the processes lightning (0.7 to 0.14) as well as avalanche (1.63 to 0.29), and to a slightly lesser extent for the processes flood (0.41 to 0.18), rockfall (0.31 to 0.10) and windstorm (0.30 to 0.19).

#### 4.2.2 Monthly distribution of fatalities

The monthly distribution of natural hazard fatalities from 1946 to 2015 showed two distinct peaks, one in summer and one in winter (Figure 6). The first peak was due to the seasonal distribution of classic "summer processes" such as lightning, floods and, to a lesser extent, also landslides and rockfall incidents, which occur most frequently in June, July and August. Additionally, the catastrophic ice avalanche event in 1965 contributed considerably to the establishment of August as the month with most loss of life (207 fatalities). The winter peak was largely caused by avalanches, the most fatal process type in Switzerland, which accounted for 242 of the overall 280 deaths in the months of January (135 avalanche related deaths out of 145 total deaths) and February (105 avalanche deaths/135 total deaths). The months of March, April and May in spring and September through December in autumn and early winter exhibited relatively low fatality numbers, i.e. below 60. The month with the fewest deaths related to natural hazard processes was November, when a total of 22 deaths occurred during the last 70 years. This corresponds to a fatality rate roughly one order of magnitude smaller than the rate in August.

These seasonal patterns resulted in a high percentages of fatalities in summer (June, July and Aug.; 41.7%) and winter (Dec., Jan. and Feb.; 32.2%). In contrast, spring (Mar., Apr. and May; 16.0%) and autumn (Sept., Oct. and Nov.; 10.1%) displayed low percentages. While 85.4% of lightning deaths and 67.7% of flood deaths occurred in summer, 70.6% of avalanche victims were killed in winter and 24.9% in spring. In autumn, only fatalities by landslides show an relatively high percentage (39.2%).

#### 4.2.3 Natural hazard fatalities classified by time of day

Based on information on the fatality event time, we assigned the cases to four different time periods: morning (06:00-11:59 local standard time), afternoon (12:00-17:59), evening (18:00-23:59) and night (00:00-05:59). Most of the fatalities occurred in the afternoon (39%), followed by the evening (23%), the morning (17%) and the night (11%). For ten percent of the fatalities, no exact event time could be determined (Figure 7). Fatalities due to lightning strikes, floods or windstorms occurred mostly in the afternoon and in the evening, whereas no characteristic time period could be distinguished for the occurrence of fatalities due to avalanches, rockfall events and landslides. However, avalanches and landslide processes showed a considerably higher percentage of deaths during the night (roughly 20%) compared to the other processes (less than 10%).

### 4.3 Spatial distribution of natural hazard fatalities in Switzerland

Fatalities caused by natural hazard processes were relatively homogenously distributed over the entire territory of Switzerland (Figure 8a). There are a few, mostly small, areas where very few or even no deaths. Multi-fatality events were much more frequent in the mountainous parts of the country (Alps) compared to the Swiss Plateau and the hilly Jura.

As can be expected, fatalities resulting from avalanches occurred mainly in the high-alpine parts of Switzerland. A few accidents were reported from the western and central Prealps (transitional areas to the high-alpine part of Switzerland). In contrast, no fatal incidents relating to avalanches occurred in the hilly Jura and in the Swiss Plateau region. Clusters are present, e.g. in the area around Andermatt in the southern part of the canton Uri (UR) and around Davos in the canton Grisons (GR). Also, a notably large number of multi-fatality avalanche events is visible in Figure 8a (see also section 5.2). Interestingly, some mountainous but populated regions showed very few or no deaths by avalanches at all. Such areas are located in several parts of Grisons (GR) and Valais (VS), as well as in parts of north-eastern Ticino (TI).

Most of the fatalities caused by landslide processes were recorded in the Alps and Prealps, and a few were documented in the Swiss Plateau. The largest number of landslide deaths occurred in the canton Valais (VS), and the worst landslide event in the database (with 13 fatalities) occurred in Gondo (VS) in October 2000. Similar to landslide accidents, fatal rockfall accidents predominantly took place in the central Alps and Prealps, e.g. in the cantons of Valais, Grisons, Vaud and Uri. A limited number of these fatalities occurred in the Jura and the Swiss Plateau.

Flood fatalities were rather homogenously distributed over Switzerland and occurred in almost all regions/cantons during the last 70 years. Still, there were considerably more cases (some of them multi-fatality incidents) in the Swiss Plateau than in any other Swiss region. The fewest deaths by floods occurred in the region Jura and the canton Valais. Some areas, such as the canton of Neuchâtel, the eastern part of canton Grisons and south-eastern Valais exhibited no flood fatalities at all.

Lightning fatalities occurred all over Switzerland, and by far most of them took place on the Swiss Plateau. However, some high-mountain areas in several different parts of Switzerland exhibited very few lightning related deaths (e.g. in cantons Grisons, Valais, Uri and Ticino). Windstorm related fatalities were also mainly registered along the Swiss Plateau. This region exhibited more than half of all windstorm related fatalities. Some cases occurred in the Jura, the Prealps and on the border to the Alps. In contrast, very few wind related fatalities occurred in the large alpine cantons Grisons, Valais and Ticino. Areas around lakes (Lake Zurich, Lake Neuchatel, Lake Geneva, Lake Constance) had clusters of windstorm fatalities due to a considerable number of capsizing accidents.

The spatial distribution of natural hazard fatalities for the different process types was confirmed by the altitude data for each event. For roughly three quarters of the fatalities, we were able to define the exact altitude at which the victim died (for slightly less than 25% of the fatalities the accident altitude was estimated). While median altitude was highest for avalanche and rockfall victims (1467 and 1082 m a.s.l., respectively), landslide and lightning victims were killed at intermediate altitudes (820 and 692 m a.s.l., respectively). In contrast, flood and windstorm fatalities were mainly registered at low altitudes representative for the Swiss Plateau (median values of 559 and 431 m a.s.l., respectively).

Figure 8b shows the overall number of fatalities per Swiss canton. Most deaths occurred in the canton of Valais (VS, 272), followed by Grisons (GR, 193), Bern (BE, 100), Ticino (TI, 71), Uri (UR, 60), Vaud (VD, 44) and St. Gallen (SG, 43). Normalizing the number of deaths to canton population (deaths per year and per one million persons) shows that cantons in the Alps with a medium to very low population density had especially higher fatality values. The relatively small and sparsely populated canton of Uri showed the highest normalized value (UR, 27.4), followed by Valais (VS, 19.1), Grisons (GR, 18.2), Appenzell Innerhoden (AI, 8.7), Obwalden (OW, 5.8), Nidwalden (NW, 5.7), Ticino (TI, 4.7) and Glarus (GL, 4.6). At the other end of this list were the urban cantons of Basel-Stadt (BL, 0.08) and Geneva (GE, 0.26) located in the regions Jura and Swiss Plateau.

### 4.4 Natural hazard fatalities classified by age and gender

The age of victims was provided for more than 93% of the natural hazard fatalities reported in our database (954 out of 1023). Overall, the age groups with the highest death toll were the 20-29 year-old (172 fatalities) and 30-39 year-old (177 fatalities) groups, followed by the 40-49 year-old group with 151 fatalities (Table 2). These values correspond to a combined 500 victims (or 48.9%) between 20 and 49 years of age. While children and teenagers (0-19) accounted for 20.9% of natural hazard deaths, people above 60 years of age made up only 12.6%.

When focusing on age related patterns for the individual process types, several discrepancies from the overall numbers attract attention. For example, about one fifth of all flood victims were younger than ten years of age, compared to 7.5% for all hazard processes (Table 2). In addition, the percentage of victims over 60 years of age was more than twice as high for flood fatalities (25.8%) as for all fatalities. Lightning rarely killed young children (2.4% in the 0-9 year-old age group) but seems to particularly affect teenagers. The 10-19 year-old age group constituted 23.2% of all lightning victims, which is much higher than the value for all processes (13.4%). Finally, windstorm victims below 20 years of age were underrepresented (10.5%) compared to the full data set (20.9%).

Gender was provided for practically all (99.5%) natural hazard fatalities in our database. Summarized for all process types, more than three quarters of all natural hazard victims were males, indicating that males

were approximately three times as likely to become fatality victims as females (Table 1). Male fatalities greatly outnumbered female deaths for every process category (percentage of male fatalities between 74.2 and 79.3%), with the exception of landslides and processes in the *other* category (including ice avalanches). While the proportions of male and female victims of landslides were quite similar (55.4 and 44.6%, respectively; Table 1), the victims killed by the ice avalanche of 1965, which destroyed a dam construction site, were practically all males (96.6%). Expressed differently, female landslide deaths represented 13.6% of all female natural hazard fatalities, whereas the same value for males was only 5.3%. Correspondingly, processes classified as *other* were the cause of death in 11.0% of male victims in our records, while this value was negligible for female victims (2.9%).

Regarding the ages of victims, the number of male victims in natural hazard fatalities was considerably larger than the number of female fatalities within every age group (Table 2). This pattern is accentuated for young adults between 20 and 39 years of age, where the percentage of male victims was more than 80%, whereas for young children from 0 to 9 years of age the percentage of female victims was highest at 34%. Accordingly, the age distribution of male and female victims of natural hazard processes exhibits a few differences. Whereas 26.4% of female fatalities were younger than 20 years of age, this applied to only 19.4% of male victims. In contrast, the percentage of young-adult victims between 20 and 39 years of age was much higher for males (37.2% of all deaths) than for females (25.2%).

### 4.5 Natural hazard fatalities in different accident circumstances

Accident circumstances, such as *activity* (work or leisure time), *locality* (on transportation routes, in open terrain, in or around buildings, on a lake, in the immediate vicinity of a stream channel) and *mode of transportation* (on foot, in a vehicle, on a boat, on skis, on a bicycle, in public transport) were analysed for the 1023 entries in our database (Table 3). Regarding the victims' activity, we found that 52% of all fatalities occurred during leisure time, 35% occurred during work, and the activity could not be assigned in 13% of the cases. The assessment of incident locality revealed that most of the fatalities occurred on transportation routes (33%), in open terrain (14%) or in or around buildings (home 20%, other buildings 6%, around buildings 3%). For all fatalities except those in buildings, the mode of transport was determined: 62% of victims were killed whilst on foot, 18% in a vehicle, 7% on a boat, 6% on skis and 1% or less in public transport or on a bicycle.

Avalanches killed more than twice as many people during leisure time (245) as during work (110). People killed by avalanches during work were mostly located on transportation routes (72), mainly travelling by foot (35) or in vehicles (16). People who died in avalanches during leisure time were usually at home (159) or were travelling on transportation routes (83). Similarly, for floods, windstorms, landslides and rockfall events, fatalities that occurred during leisure time were at least twice as frequent as those that occurred during work. The single hazard event with the largest number of people killed during work was the ice

avalanche in Mattmark with 88 deaths (category *other*). People fatally struck by lightning were working in 43% of the cases (Table 3).

Most victims of floods were killed in the stream channel sector (63) and were usually carried along by the high water. Flood related fatalities on transportation routes were also relatively common (31), with 11 victims travelling in vehicles, 16 by foot, two by bicycle, one in public transport and one case unclear. Regarding windstorm related incidents, most of the fatalities occurred on lakes with victims located in boats (44). Thirty-four windstorm related fatalities occurred on transportation routes, where 20 victims were killed in vehicles, 13 on foot and one on a bicycle. Loss of life caused by landslide processes occurred mostly in and around buildings (40). Twenty people were killed by landslides whilst on transportation routes, and 11 of these victims were in vehicles, eight on foot and one in public transport. Most people killed by rockfall were on transportation routes (58), of which 30 people were travelling by foot and 27 in a vehicle (one unclear), whereas 14 people were killed by rockfall on open terrain. Most of the lightning fatalities occurred on open terrain (95 victims, of which 90 were moving by foot), in or around buildings (33) or on transportation routes (28 victims, of which 13 were travelling by foot, 10 in or on a vehicle and 5 by bicycle).

# 5. Discussion

## 5.1 Data quality

### 5.1.1 Completeness of the data set

Several factors influenced the integrity of our data set: (i) Fatal accidents caused by natural hazard processes during the 70-year study period could simply not have been reported in the Neue Zürcher Zeitung (NZZ) or in the additional data sources used. (ii) The NZZ is a German written newspaper. The coverage of the news includes all parts of Switzerland (e.g. regarding severe natural hazard events), but there might be a bias towards an underreporting in the other language regions (French, Italian, Rhaeto-Romanic), which represent roughly one third of the population. (iii) For practical reasons, we had to apply a limited number of keywords in our search. Thus, we could have missed reports on natural hazard fatalities in the data sources because our keywords were not used in the respective news coverage. (iv) There were a few small data gaps in our main data source. Some issues of the NZZ were not present in the data portal applied; hence, we possibly missed fatalities that occurred during such periods. (v) The NZZ archive was established by scanning all newspaper issues and applying a character recognition program (with the exception of all new editions available digitally since 1994). This procedure is susceptible to errors whenever the character recognition fails due to insufficient scan quality. We encountered this problem when searching for keywords in the first few decades of the study period, mainly for long keywords. (vi) Finally, media articles sometimes mix up technical terms and thus hazard processes (a problem already discussed in Badoux et al., 2014). For example, we suspect that several fatal debris flows were not exactly

identified as such in the media during the first few decades of the study period because this process was not yet commonly understood.

All in all, we acknowledge that our database for the time period between 1946 and 2015 is not fully complete and that we might have missed a certain number of natural hazard deaths, mainly in the first decades of our study period. Taking into account our success rate for the validation period (see section 3.3), we are confident that we passed over considerably less than 10% of the fatalities that occurred. The main reason for this is that graver events with several deaths were almost certainly reported in the NZZ. Hence, we believe that our database is valuable in its present state.

### 5.1.2 Quality of the data

The quality of our natural hazard fatality data was assessed by considering the levels of uncertainty associated with the incident characteristics and circumstances. In general, uncertainty was low (Table 4). This finding applies primarily to the date of events, the victims' gender and age, and the locality of the incident (data was labelled as *certain* in 88% or more of the cases). Exact reporting of the time an incident occurred was often lacking, and we had to estimate the event time for more than one third of the fatalities. Data was unavailable in more than 10% of the cases only for the variables event time and victims' activity. Finally, data quality increased in the course of our study period, in that all variables except one (victims' activity) show lower uncertainty for the sub-period between 1981 and 2015.

## 5.2 The impact of multi-fatality events

A total of 635 natural hazard events with fatal consequences occurred in Switzerland during the study period (Table 5). The most common situation observed in the last 70 years was for one victim to be killed in a natural hazard incident. Roughly 50% of the total number of victims lost their life in one of the 507 single-fatality events recorded in the database. While incidents with two or three fatalities occurred 73 and 25 times, respectively, 25 incidents registered four to ten fatalities. Only five events caused more than ten deaths each, resulting in a combined 15.7% of the total number of victims in the entire study period (Table 5).

The largest of those five events was the Mattmark ice avalanche of 1965 that killed 88 people. The second and third worst multi-fatality events were both avalanches. The first of these avalanches occurred in February 1970 in Reckingen (canton of Valais) and represents one of the largest avalanche disasters in the Alps of the 20th century (30 deaths). Amongst other damage, the avalanche destroyed a Swiss army barrack and residential buildings, killing 19 officers and 11 civilians. The latter incident with 19 deaths occurred in January 1951 in Vals (canton of Grisons) during the worst avalanche winter in Swiss history (Figures 4 and 5).

Regarding the relative importance of single-fatality events, we detected two distinct categories of natural hazard process types. Flood, rockfall, windstorm and lightning incidents with one victim were responsible for more than 70% of the total fatalities associated with each of these process types. This value was much smaller for avalanches (26.2%) and landslides (31.1%) (Table 5). This distinction is probably related to the fact that fatal victims of both avalanche and landslide process types are more often killed in buildings (around 50% of all cases) than victims of other hazard processes. Affected buildings are likely to be occupied or inhabited by more than one person, and thus, multi-fatality events are more frequent for avalanches and landslides. Lightning was associated with the highest percentage of deaths in single-fatality events in Switzerland (87.8%). This value is only slightly lower than percentages reported for the USA (90.9%, Curran et al., 2000) and Australia (92%, Coates et al., 1993).

### 5.3 Evolution of fatality numbers during the last 70 years in Switzerland

General attempts to explain the decrease in natural hazard fatalities observed in other countries and globally have been made previously. For example, Goklany (2007) indicated that societies' collective adaptive capacities lead to reduced fatality rates due to extreme weather. For the US, Curran et al. (2000) attributed a decline in lightning related deaths over time to improved medical care, emergency communication and transportation, as well as better awareness of the serious threat posed by lightning. Improved forecasting, process detection and warning systems may have also contributed to the decrease in fatalities (Curran et al., 2000; López and Holle, 1996). Furthermore, fewer people currently work in open fields than in the past, and the expansion of urban areas has provided more buildings and structures to attract lightning away from people (Elsom, 2001).

A significant reduction in avalanche fatalities in settlements and on transportation corridors since the 1970's has been noted throughout the European Alps (e.g. Techel et al., in prep.), as well as in North America (e.g. Page et al., 1999; Jamieson et al. 2010). Explanations for this decrease include (i) large investments into permanent avalanche defence structures, which reduce the potential for catastrophic avalanches (e.g. SLF, 2000); (ii) hazard mapping and risk assessment to evaluate appropriate measures for the protection or closure of avalanche-threatened sections of roads during winter (e.g. Wilhelm, 1997; Margreth et al., 2003); (iii) the preventive artificial release of avalanches (e.g. Stoffel, 2001); (iv) the improved avalanche education of local authorities responsible for avalanche safety (e.g. Bründl et al., 2004); and (v) an improved and more widely distributed avalanche forecast (Etter et al., 2008).

Many fatalities related to floods and inundations occur because people act imprudently and put themselves in dangerous situations. We assume that about half of all flood deaths can be ascribed to such inappropriate behaviour, for example when victims are carried away by floodwaters or surprised in their home by rapidly intruding surface water (see also section 5.6). Hence, although Swiss flood protection has been developed and refined considerably in the last decades (e.g. BWG, 2001), implying large financial investments, fatal

events can only be prevented by such measures to a limited extent. This could partly explain the relatively moderate decrease in flood fatality over our study period. However, it has to be taken into consideration that deaths caused by floods were dramatically reduced since the middle of the 19th century in Switzerland, as showed e.g. in Petrascheck (1989). Still, it will be very important in the future to better inform and train people who regularly stay in or close to flood prone areas. A similar recommendation could be made for windstorms, as fatalities related to these events also often occurred due to negligent behaviour in dangerous situations.

**5.4 Demography of fatality data**

Our study revealed that male natural hazard fatalities have been much more frequent than female fatalities (75.9% male vs. 23.7% female). This pattern has also been observed in other countries, for example for fatal flood (Ashley and Ashley, 2008; Coates, 1999) and lightning (Singh and Singh, 2015; Navarrete-Aldana et al., 2014; Elsom, 2001; Curran et al., 2000) victims. To explain this striking gender difference, we first need to focus on work related deaths, which represent slightly more than one third of all recorded fatalities (Table 3). A remarkable 93.5% of all natural hazard victims killed at work were men, and this percentage was at least 95% in the four first decades of the study period. This finding is probably mainly due to the fact that (i) during the study period, the ratio of men working full time was considerably higher than the ratio of women working full time, especially in the first half of the period; and (ii) many accidents were associated with occupations that are physically demanding and thus were, and to some extent still are, almost exclusively carried out by men, such as farm labour, forestry work, construction work, road maintenance (e.g. snow clearing) and rescue services. Thus, men have been more involved than women in employment (often involving outdoor work) that puts them at risk of dying in a natural hazard event.

More than half of all fatalities in our data set occurred during leisure time (Table 3). Of these 536 victims, approximately two thirds were men and one third were women. Obviously, the predominance of male fatalities was not as strong as that observed in work related accidents, but it was still considerable. We assume that the gender difference observed for fatalities during leisure time activities was due to the higher risk perception of women compared to men (e.g. Bubeck et al., 2012; Lindell and Hwang, 2008; Tekely-Yesil et al., 2011). This effect probably leads to a more cautious and less adventurous behaviour of women, which could explain the difference between the number of male and female fatalities. What applies for work might also be valid for recreational activities and it seems that men have a greater disposition towards risk-taking than women. However, a growing recognition of the equality of the genders has led to a greater proportion of women being at risk relative to men (Coates, 1999). This trend is somewhat supported by our data, in that 55.6% of all leisure-time fatalities in the last decade of the study period involved women.

### 5.5 Effects of location, mode of transport, activity and inappropriate behaviour

The percentage of people who drowned in vehicles was much smaller in Switzerland (17% of all flood victims) compared to values reported in studies of other countries. For example, 77% of flood fatalities were vehicle related in a study about the US state of Texas (Sharif et al., 2015), and 63% were vehicle related in a study of the US as a whole (Ashley and Ashley, 2008). It was suggested that this large number of fatalities occurs because people incautiously try to cross rivers in their vehicles (Sharif et al., 2015). Most of the people in our study were killed while travelling on foot (62%). This percentage includes people who accidentally fell, were swept into the floodwaters, or tried to walk through the floodwaters. We assume that most of these fatalities occurred because of inappropriate behaviour and underestimation of risk and thus could have been prevented. The same could be said for all the lightning deaths in our study. In most cases, the danger was underrated (e.g. people were struck during work on open fields) or people used inadequate shelter (e.g. under trees). For fatalities due to lightning, a clear trend over time away from outdoor worker casualties towards recreational accidents was apparent in our data set and has also been observed elsewhere (López et al., 1995; Coates et al., 1993). Inappropriate behaviour also led to many of the deaths related to windstorm events. Here, most fatalities occurred on lakes due to capsized boats, when the victim(s) underestimated the heavy winds. Further, many victims were struck by falling tree (parts) or other material that was swirling through the air. Regarding avalanche incidents, , most of the fatalities in our database occurred in buildings (185, 49%) because we excluded deaths due to sports like ski touring and free skiing. We assume victims killed in buildings thought they were safe. Similarly, in the landslide process category, most of the fatalities occurred in buildings (54%), suggesting that the inhabitants were surprised and unable to react to the threat.

### 5.6 Comparison of recent Swiss natural hazard fatality data with historic data

In our data set from Switzerland, less than 1% of all events caused more than 15% of the overall fatalities, and since 1946 no single natural hazard incident killed more than 88 people (section 5.2). Compared to some catastrophic historical events that occurred on Swiss territory, 88 victims is considerable but not exceedingly great. In this section, we briefly address three events that probably represent the most devastating natural hazard incidents of the last 1000 years in Switzerland: an earthquake, a rockslide, and a rockslide/flood wave combination.

On 18 October 1356 in the region of Basel, an earthquake occurred that is regarded as the strongest historically documented earthquake event in central Europe (Swiss Seismological Service, 2004). The inferred magnitude of the seism lies between 6.5 and 7.0. This event destroyed large parts of Basel, but the overall number of victims remains very uncertain (Fäh et al., 2009) and contemporary sources are sparse. While 300 deaths were associated with this incident in a chronicle from around 1580, miscellanies of the 16th century report a maximum figure of 2000 fatalities. However, , the latter number seems highly

exaggerated (Fäh et al., 2009). If an identical earthquake occurred today, experts fear it would cause approximately1500 deaths (Katarisk, www.katarisk.ch).

Approximately 160 years later, during an intense rainfall period in September 1513, a massive rockslide 10 to 20 million $m^3$ in volume failed from Pizzo Magno and blocked the river Brenno at the end of the Blenio valley (canton of Ticino) just upstream of the village of Biasca (Heim, 1932). Due to this damming of the Brenno, a shallow lake with a final length of 5 km was formed and two small villages were submerged. Roughly 20 months after the rockslide, in May 1515, the debris plug eroded and eventually collapsed. A massive surge of debris and water flooded Biasca and swept down the valley of the Ticino to the town of Bellinzona and further to Lago Maggiore (Eisbacher and Clague, 1984; Bonnard, 2011). Approximately 600 people were killed and some 400 buildings were destroyed.

On 2 September 1806, approximately 20 million $m^3$ of material (Eisbacher and Clague, 1984; other sources cite volumes up to 40 million $m^3$) failed on the southern slopes of Rossberg mountain in central Switzerland and subsequently destroyed and buried the villages of Goldau and Röthen, as well as parts of two adjacent hamlets. This so-called Goldau rock avalanche had a runout of about 1000 m and caused the destruction of more than 100 houses, 220 stables and barns, two churches and two chapels, and it killed 457 people. Several signs (e.g. tension crack formation and rockfall) predicted a failure several years before the event, but most people did not heed the warning and remained in the hazardous area (Heim, 1932; Eisbacher and Clague, 1984).

Further rockslides with catastrophic consequences occurred in the last centuries, such as an event in Elm in 1881 and an event in Plurs/Piuro (formerly Swiss territory, now Italy) in 1618. The latter event was exceptionally devastating and probably killed more than 2000 people. However, both events were to a large extent man-made, as intensive mining activity crucially destabilized the slopes that ultimately failed. Catastrophic events like the ones addressed above that seem to occur every 100 to 300 years in Switzerland and cause hundreds of victims would greatly change the fatality rates described in this contribution for the worse.

### 5.7 Comparison of Swiss natural hazard fatality data with data from abroad

The EM-DAT International Disaster Database is a free and searchable source of information on world-wide victims of natural disasters (EM-DAT, www.emdat.be). In that database, we found 21 database entries for Switzerland in our study period, and 329 fatalities associated with natural hazard processes that were considered in our study. Avalanche events led to most of the fatalities recorded in EM-DAT (275), followed by storms (24), landslides (20) and riverine floods (10). A considerable difference from our overall results exists because EM-DAT focuses only on large catastrophic events with at least ten fatalities,

100 affected people, a call for international assistance, or the declaration of a state of emergency. This leads to an underestimation of total fatalities, a problem also stated by Petley (2012).

Worldwide, EM-DAT registered a total of 5 million fatalities, including those due to earthquake (1.51 million, 30.1%), mass movement (0.06 million, 1.2%; including landslide, avalanche, rockfall and subsidence), flood (2.48 million, 49.2%) and storm (0.98 million, 19.5%) for the period of 1946 to 2015. Normalized by population (UN, Department of Economic and Social Affairs, Population Division, World Population Prospects, the 2015 Revision, http://esa.un.org), this results in a mean yearly fatality rate of 19 victims per million people (earthquake 4.6, flood 11.4, mass movement 0.2, storm 3.2). This rate is much higher compared to the average annual fatality rate determined in our study for Switzerland (overall 2.5 fatalities per million population and year; flood 0.29, landslide 0.16, rockfall 0.20, storm 0.24, lightning 0.42, avalanche 0.96, other 0.23; Table 1).

With the different data collection methods in mind, explanations for these differences could be that most events that occurred in Switzerland during our study period did not cause more than ten fatalities (all but five events, Table 5). Due to the geographical location of the study area, some very deadly hazard processes, such as tropical storms, did not occur. Further, even though severe earthquakes are possible in Switzerland and have been observed historically (see section 5.6), only three such fatalities (all in 1946) were registered. Further, Switzerland exhibits a considerable network of protection measures and a well-developed risk management system (see section 5.3). These tactics probably additionally help to keep fatality rates low in comparison to other regions of the world, particularly less privileged ones (e.g. Bründl et al., 2004; Lateltin et al., 2005).

Higher flood fatality rates compared to Switzerland have been reported for Texas (1.08 fatalities per million population per year) as well as several other US states (Sharif et al., 2015) and also for India (1.5; Singh and Kumar, 2013). Coates (1999) found a large decrease in the flood fatality rate in Australia, from 239.8 in the 1800's to 0.4 in the 1990's. Also, a decreasing frequency of fatalities due to landslides was found in Italy, with 1.8 fatalities per million population per year in 1950 and 1.4 in 1999 (Guzetti, 2000). The average annual rate of fatalities due to lightning we found for Switzerland was higher than that reported for India (0.25; Singh and Singh, 2015) and similar to that reported for the USA (0.42; Curran et al., 2000). The decrease in lightning related fatalities we found over the last 70 years has also been observed in the USA (López and Holle, 1996) and in England and Wales (Elsom, 2001). In a comparative global summary of published lightning fatality estimates, Singh and Singh (2015) listed a value for 19th century Switzerland of 6.0 deaths per million population per year, which is more than ten times the value estimated in our study. Even though this figure was based on only two years of data (1876-1877), it illustrates the various improvements (e.g. better medical care, increased awareness) and the socio-economic changes that have occurred in Switzerland over the past 150 years.

### 5.8 Comparison of natural hazard fatality data with data from other accidental deaths

In his recent contribution, Goklany (2007) assessed death and death rates due to extreme weather events, such as extreme heat, extreme cold, floods, lightning, tornados and hurricanes. This analysis indicated that globally, as well as for the US, the aggregate contribution of extreme weather events to overall mortality is relatively small, ranging from 0.03% (globally) to 0.06% (US).More specifically, the global contribution of fatalities due to extreme weather events to all accidental deaths is also quite small (0.4%). For example, based on EM-DAT data for 2000-2006 and World Health Organization data for 2002, Goklany (2007) showed that, while roughly 20,000 people per year die during extreme weather events, many more are killed in road traffic accidents (approx. 1.2 million people in 2002).

The natural hazard processes considered in the study mentioned above are not exactly the same as in the present study (e.g. heat waves were not considered here and mass movements probably were not included in Goklany, 2007) and the two study periods differ. Nevertheless, Goklany's point also applies to our Swiss analysis: fatality and fatality rates from the different hazard processes studied in the present paper are relatively low. A comparison of our data with Swiss road and railroad accident fatality data (Swiss Statistics, www.bfs.admin.ch) confirms the ratio suggested above at the global level. For the same 70-year time frame from 1946 to 2015, a total of 64,561 people died on Swiss roads during traffic accidents. This corresponds to an average value of 922 fatalities per year, with a maximum of 1773 deaths in 1971 and a minimum of 243 deaths in 2014. Thus, on average for our study period, more than sixty times as many people were killed in traffic accidents than by natural hazards. Swiss railroad fatalities also clearly outnumbered natural hazard related fatalities, although to a lesser extent than traffic fatalities (4871 victims from 1946 to 2014, corresponding to roughly five times the number of natural hazard related deaths). Note that both road and railroad fatality numbers show a distinct decrease in Switzerland since the 1970s, similar to the natural hazard fatality data presented here (Figs. 3 and 5).

Comparing the presented fatality numbers with those of recreational accidents in mountainous terrain, which were not considered in our study, shows similar patterns (see section 3.4). For instance, data from the Swiss Alpine Club (SAC, www.sac-cas.ch) show that mountaineering accidents (e.g. mountaineering, hiking, climbing, canyoning) caused by lightning and rockfall led to 62 fatalities from 2000 to 2013, compared to 21 fatalities in our database for the same two processes (with six fatalities appearing in both data sources). Comparable patterns exist for avalanche fatalities: during the last twenty years (1995/96-2014/15), 15 times more people lost their life during recreational activities in unsecured backcountry terrain than in settlements or on transportation corridors in Switzerland (Techel et al., 2015). On a larger spatial scale covering the entire European Alps, this proportion is even more pronounced (a factor of approximately 30 for the 15 years from 2000/01 to 2014/15; Techel et al., in prep.). In the vast majority of cases, victims in unsecured terrain triggered the avalanche themselves (e.g. Schweizer and Lütschg, 2001), as opposed to fatalities caused by avalanches in settlements and on transportation routes, where most

avalanches released naturally (85% of the avalanche victims reported in the present study were killed by naturally triggered events).

**6. Conclusions**

In this study, we compiled data from the Swiss flood and landslide database, the destructive avalanche database, and information collected in an in-depth newspaper search to establish a new database of fatalities caused by natural hazard processes in Switzerland. For a 70-year period from 1946 to 2015, we were able to assemble detailed data on 635 events during which 1023 people were killed by processes we summarized into seven hazard types (flood, landslide, rockfall, lightning, windstorm, avalanche, and *other* processes). Fatalities that occurred in connection with high-risk sports or certain popular summer and winter outdoor sport activities were not considered. Snow avalanches claimed 378 victims and clearly represent the deadliest natural hazard process in Switzerland. With 164 deaths, less than half as many fatalities were caused by lightning. Floods, windstorms, rockfall events and landslides killed 124, 105, 85 and 74 people, respectively.

Natural hazard fatalities were considerably variable over time. The number of people who died in one single year varied by one to two orders of magnitude. In 1951 and 1965, fatal events resulted in 112 and 108 deaths, respectively, whereas five years had only two fatalities each. For the 70-year study period, the average and median number of people who died due to natural hazards amounted to 14.6 and 9.0, respectively. Annual loss of life data showed a decrease over time, which was primarily induced by a marked decrease in deaths due to avalanches and lighting strikes. The reduction in avalanche fatalities in settlements and on transportation routes is a trend that has also been observed in other European countries and elsewhere. The decrease is due to improvements in both technical (defence structures, preventive artificial release) and organizational (e.g. hazard mapping, emergency planning) measures and to significant progress in avalanche education and forecasting. A reduction in lightning victims during the last century also occurred in many other countries around the world. In Switzerland, the main reasons for this distinct decrease might be that today fewer people work outdoors and there is a much improved awareness of the threat posed by lightning.

Most people were killed by natural hazard events in summer (JJA, 41.7% of fatalities) and winter (DJF, 32.2%). Accordingly, the four months with the largest number of victims were August (20.2%), January (14.2%), February (13.2%) and July (12.9%). While the summer peak mainly occurred due to flood and lightning events (together with one catastrophic ice avalanche in August 1965), the winter peak was caused by snow avalanche incidents. Furthermore, almost two thirds of the fatalities took place in the afternoon and evening.

Natural hazard fatalities were quite homogenously distributed over Switzerland. However, mountainous parts of the country (Prealps, Alps) were somewhat more prone to fatal events compared to the Swiss Plateau and the Jura. The reason for this is that avalanche fatalities, and to a slightly lesser extent rockfall and landslide fatalities, occur mainly in the alpine parts of Switzerland. In contrast, deadly events associated with floods and lightning were observed in practically all regions of Switzerland, with a definite maximum occurring along the Swiss Plateau. Finally, windstorm related fatalities were mostly observed on the Swiss Plateau, especially on lakes.

The age groups with the largest number of natural hazard victims were the 20-29 and 30-39 year-old groups (172 and 177, respectively), and almost 50% of all victims were between 20 and 49 years of age. Young children (0-9 years of age) were the age group that was most underrepresented in the fatality data compared to the Swiss age distribution. Three quarters of all fatalities were men, and men outnumbered women in all process types except landslides. We assume that this large gender difference was strongly influenced by two factors: first and probably more importantly, virtually all work related fatality incidents involved physically difficult occupations that put workers at risk and where the majority of workers are men; second, we speculate that women have a considerably higher risk perception compared to men and that, accordingly, men have a greater disposition towards risk taking than women. Both of these points are (probably) especially relevant for the first half of the study period.

Large catastrophic events with several hundreds of fatalities have occurred in Switzerland in the last 1000 years (e.g. Basel earthquake of 1356, Goldau rock avalanche of 1806), but not during the period studied here. Apart from the tragic Mattmark ice avalanche of 1965 that killed 88 people, only one landslide and three avalanche events caused more than ten victims in the period from 1946 to 2015. Together, these five incidents led to 161 deaths, which correspond to 15.7% of all fatalities in the data set. Hence, single-fatality events (that account for approximately half of the total natural hazard deaths) and events with up to a few victims strongly influence the natural hazard fatality statistics in the recent past. For some process types, such as lightning and rockfall, this influence is particularly strong. When compared with natural hazard fatality rates in other countries or with accidental deaths from other causes in Switzerland, the fatality numbers presented here are quite low. For example, traffic accidents kill an average of approximately 60 times more people than natural hazard events. Nevertheless, we think that current annual natural hazard fatality numbers can still be further reduced by investing in and further developing both structural and organizational (e.g. alarm systems, emergency planning, hazard awareness creation) protection measures. The data set (and analysis) presented here can be used by decision makers at different political levels (municipal, cantonal and federal authorities) to plan and implement such measures.

## Acknowledgements

We are very grateful to G. Antoniazza, who helped screen many newspaper articles to establish this database. We thank J. Keel, of the Swiss Media Database for excellent support throughout our newspaper search. Furthermore, the authors would like to thank D. Rickenmann, B. McArdell, C. Berger, C. Rickli, E. Maidl, M. Buchecker (all at WSL), U. Mosimann (Swiss Alpine Club), and F. Haslinger (Swiss Seismological Service) for insightful discussions and K. Liechti (WSL) for support with data analysis. Finally, we are grateful to staff at the Federal Office for the Environment (FOEN), especially R. Loat and G. R. Bezzola, for their considerable contribution to the maintenance of the Swiss flood and landslide damage database. M. Dawes is acknowledged for improving the quality of the manuscript.

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

Table 1: Natural hazard fatalities in Switzerland (1946-2015) classified by gender.

| Process type | Total fatalities | | Normalized fatalities | Female fatalities | | Male fatalities | | Gender unclear | Percent female deaths | Percent male deaths | Altitude (A) |
|---|---|---|---|---|---|---|---|---|---|---|---|
| | [deaths] | [%] | [deaths $10^{-6}$ $a^{-1}$] | [deaths] | [%] | [deaths] | [%] | [deaths] | [%] | [%] | [m a.s.l.] |
| **Flood** | 124 | 12.1 | 0.29 | 31 | 12.8 | 92 | 11.9 | 1 | 25.0 | 74.2 | 559 |
| **Landslide** | 74 | 7.2 | 0.16 | 33 | 13.6 | 41 | 5.3 | 0 | 44.6 | 55.4 | 820 |
| **Rockfall** | 85 | 8.3 | 0.20 | 21 | 8.7 | 64 | 8.2 | 0 | 24.7 | 75.3 | 1082 |
| **Windstorm** | 105 | 10.3 | 0.24 | 25 | 10.3 | 79 | 10.2 | 1 | 23.8 | 75.2 | 431 |
| **Lightning** | 164 | 16.0 | 0.42 | 33 | 13.6 | 130 | 16.8 | 1 | 20.1 | 79.3 | 692 |
| **Avalanche** | 378 | 37.0 | 0.96 | 92 | 38.0 | 285 | 36.7 | 1 | 24.3 | 75.4 | 1467 |
| **Other** | 93 | 9.1 | 0.23 | 7 | 2.9 | 85 | 11.0 | 1 | 7.5 | 91.4 | 2081 |
| **All processes** | **1023** | **100** | **2.50** | **242** | **100** | **776** | **100** | **5** | **23.7** | **75.9** | **1179** |

(A) Note that for snow avalanches only, the altitude of the lowest deposition point was used for technical reasons, which might lead to a slight underestimation in comparison with the other processes.

Table 2: Natural hazard fatalities in Switzerland (1946-2015) classified by age groups.

| Process type | Age group | 0-9 | 10-19 | 20-29 | 30-39 | 40-49 | 50-59 | 60-69 | 70-79 | 80-89 | 90-99 | Age un-known | Total |
|---|---|---|---|---|---|---|---|---|---|---|---|---|---|
| **Flood** | [deaths] | 25 | 15 | 10 | 19 | 13 | 8 | 16 | 13 | 3 | 0 | 2 | 124 |
| | [%] | *20.2* | *12.1* | *8.1* | *15.3* | *10.5* | *6.5* | *12.9* | *10.5* | *2.4* | *0* | *1.6* | *100* |
| **Landslide** | [deaths] | 1 | 13 | 6 | 14 | 13 | 10 | 4 | 7 | 1 | 0 | 5 | 74 |
| | [%] | *1.4* | *17.6* | *8.1* | *18.9* | *17.6* | *13.5* | *5.4* | *9.5* | *1.4* | *0* | *6.8* | *100* |
| **Rockfall** | [deaths] | 4 | 13 | 11 | 11 | 8 | 14 | 15 | 1 | 1 | 0 | 7 | 85 |
| | [%] | *4.7* | *15.3* | *12.9* | *12.9* | *9.4* | *16.5* | *17.6* | *1.2* | *1.2* | *0* | *8.2* | *100* |
| **Windstorm** | [deaths] | 3 | 8 | 17 | 18 | 13 | 14 | 5 | 7 | 1 | 0 | 19 | 105 |
| | [%] | *2.9* | *7.6* | *16.2* | *17.1* | *12.4* | *13.3* | *4.8* | *6.7* | *1.0* | *0* | *18.1* | *100* |
| **Lightning** | [deaths] | 4 | 38 | 29 | 25 | 24 | 18 | 12 | 7 | 1 | 0 | 6 | 164 |
| | [%] | *2.4* | *23.2* | *17.7* | *15.2* | *14.6* | *11.0* | *7.3* | *4.3* | *0.6* | *0* | *3.7* | *100* |
| **Avalanche** | [deaths] | 39 | 48 | 74 | 68 | 66 | 44 | 18 | 11 | 2 | 1 | 7 | 378 |
| | [%] | *10.3* | *12.7* | *19.6* | *18.0* | *17.5* | *11.6* | *4.8* | *2.9* | *0.5* | *0.3* | *1.7* | *100* |
| **Other** | [deaths] | 1 | 2 | 25 | 22 | 14 | 3 | 2 | 1 | 0 | 0 | 23 | 93 |
| | [%] | *1.1* | *2.2* | *26.9* | *23.7* | *15.1* | *3.2* | *2.2* | *1.1* | *0* | *0* | *24.7* | *100* |
| **All processes** | **[deaths]** | **77** | **137** | **172** | **177** | **151** | **111** | **72** | **47** | **9** | **1** | **69** | **1023** |
| | **[%]** | ***7.5*** | ***13.4*** | ***16.8*** | ***17.3*** | ***14.8*** | ***10.9*** | ***7.0*** | ***4.6*** | ***0.9*** | ***0.1*** | ***6.7*** | ***100*** |
| **Male** | [deaths] | 51 | 99 | 144 | 144 | 115 | 83 | 53 | 34 | 6 | 1 | 46 | 776 |
| | [%] | *6.6* | *12.8* | *18.6* | *18.6* | *14.8* | *10.7* | *6.8* | *4.4* | *0.8* | *0.1* | *5.9* | *100* |
| **Female** | [deaths] | 26 | 38 | 28 | 33 | 36 | 28 | 19 | 12 | 3 | 0 | 19 | 242 |
| | [%] | *10.7* | *15.7* | *11.6* | *13.6* | *14.9* | *11.6* | *7.9* | *5.0* | *1.2* | *0* | *7.9* | *100* |

Table 3: Natural hazard fatalities in Switzerland (1946-2015) classified by victims' activity, accident locality and victims' mode of transport.

| | | Flood | Landslide | Rockfall | Windstorm | Avalanche | Lightning | Other | All processes | All processes [%] |
|---|---|---|---|---|---|---|---|---|---|---|
| **Activity** | **work** | 35 | 13 | 15 | 24 | 110 | 70 | 88 | **355** | **34.7** |
| | **leisure time** | 73 | 40 | 40 | 52 | 245 | 82 | 4 | **536** | **52.4** |
| | **other/unclear** | 16 | 21 | 30 | 29 | 23 | 12 | 1 | **132** | **12.9** |
| | **total** | **124** | **74** | **85** | **105** | **378** | **164** | **93** | **1023** | **100** |
| **Locality** | **in or around a building** | 14 | 40 | 8 | 4 | 192 | 33 | 2 | **293** | **28.6** |
| | **on transportation routes** | 31 | 20 | 58 | 34 | 164 | 28 | 1 | **336** | **32.8** |
| | **in open terrain** | 4 | 11 | 14 | 9 | 4 | 95 | 1 | **138** | **13.5** |
| | **on a lake** | | | 1 | 44 | | 1 | | **46** | **4.5** |
| | **in a stream channel** | 63 | 2 | | | | | | **65** | **6.4** |
| | **other/unclear** | 12 | 1 | 4 | 14 | 18 | 7 | 89 | **145** | **14.2** |
| | **total** | **124** | **74** | **85** | **105** | **378** | **164** | **93** | **1023** | **100** |
| **Mode of transport** | **on foot** | 77 | 22 | 51 | 36 | 78 | 112 | 90 | **466** | **61.6** |
| | **in vehicle** | 20 | 11 | 27 | 20 | 46 | 14 | | **138** | **18.2** |
| | **by boat** | 5 | | | 44 | | 1 | | **50** | **6.6** |
| | **by ski** | | | | | 49 | | | **49** | **6.5** |
| | **by bicycle** | 3 | | | 1 | | 6 | | **10** | **1.3** |
| | **by public transport** | 1 | 1 | | | 4 | | | **6** | **0.8** |
| | **other/unclear** | 9 | 0 | 2 | 4 | 16 | 5 | 2 | **38** | **5.0** |
| | **total [A]** | **115** | **34** | **80** | **105** | **193** | **138** | **92** | **757** | **100** |

[A] Note that 266 fatalities that occurred in buildings are not considered in this section (mode of transportation)

Table 4: Quality of the data describing the circumstances of fatal natural hazard events and the victims (certain = information regarding this variable is fully reliable; probable = information regarding this variable was deduced but is very probable; unknown = no information available).

| | **Certain** | | **Probable** | | **Unknown** | | **Total** |
|---|---|---|---|---|---|---|---|
| | **[no. of deaths]** | **[%]** | **[no. of deaths]** | **[%]** | **[no. of deaths]** | **[%]** | **[no. of deaths]** |
| **Date of event** | 965 | 94.3 | 58 | 5.7 | 0 | 0.0 | 1023 |
| **Time of event** | 558 | 54.5 | 360 | 35.2 | 105 | 10.3 | 1023 |
| **Gender** | 1015 | 99.2 | 3 | 0.3 | 5 | 0.5 | 1023 |
| **Age** | 938 | 91.7 | 16 | 1.6 | 69 | 6.7 | 1023 |
| **Activity** | 788 | 77.0 | 107 | 10.5 | 128 | 12.5 | 1023 |
| **Locality** | 898 | 87.8 | 101 | 9.9 | 24 | 2.3 | 1023 |
| **Mode of transportation [A]** | 541 | 71.5 | 181 | 23.9 | 35 | 4.6 | 757 |

[A] Note that 266 fatalities that occurred in buildings are not considered in this row (mode of transportation)

Table 5: Fatal natural hazard events in Switzerland (1946-2015) classified by number of victims. The last two columns indicate the percentage of people that died in single fatality events (second to last column) and in events with more than ten deaths (last column); note that in the bottom row 100% corresponds to 1023 deaths for all process types.

| | **Number of events with *n* deaths** | | | | | **All fatal events** | **Percent of people killed in events where *n*=1** | **Percent of people killed in events where *n*>10** |
|---|---|---|---|---|---|---|---|---|
| | ***n*=1** | ***n*=2** | ***n*=3** | **4≤*n*≤10** | ***n*>10** | | | |
| | | | | | | | **[%]** | **[%]** |
| **Flood** | 97 | 9 | 3 | – | – | 109 | 78.2 | 0.0 |
| **Landslide** | 23 | 13 | 2 | 1 | 1 | 40 | 31.1 | 17.6 |
| **Rockfall** | 68 | 4 | – | 2 | – | 74 | 80.0 | 0.0 |
| **Windstorm** | 74 | 10 | 1 | 2 | – | 87 | 70.5 | 0.0 |
| **Lightning** | 144 | 10 | – | – | – | 154 | 87.8 | 0.0 |
| **Avalanche** | 99 | 27 | 18 | 20 | 3 | 167 | 26.2 | 15.9 |
| **Other** | 2 | – | 1 | – | 1 | 4 | 2.2 | 94.6 |
| **All processes** | **507** | **73** | **25** | **25** | **5** | **635** | **49.6** | **15.7** |

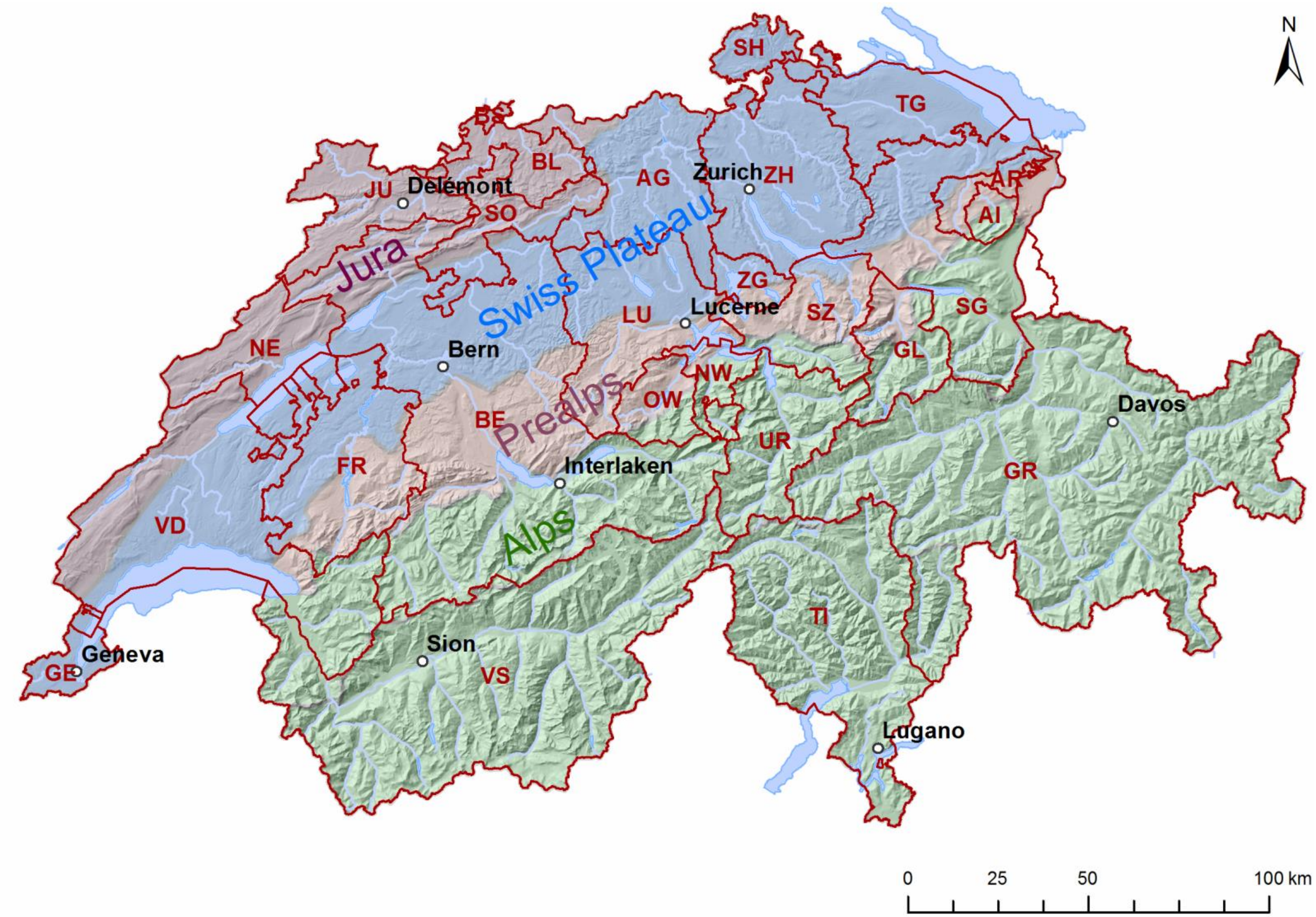

**Figure 1: Study area showing the Swiss cantons (red polygons with abbreviations, see www.bfs.admin.ch/bfs/portal/en/index/dienstleistungen/premiere_visite/03/03_02.html), the geomorphologic-climatic regions Jura, Swiss Plateau, Prealps and Alps (areas with different background colours), and key cities.**

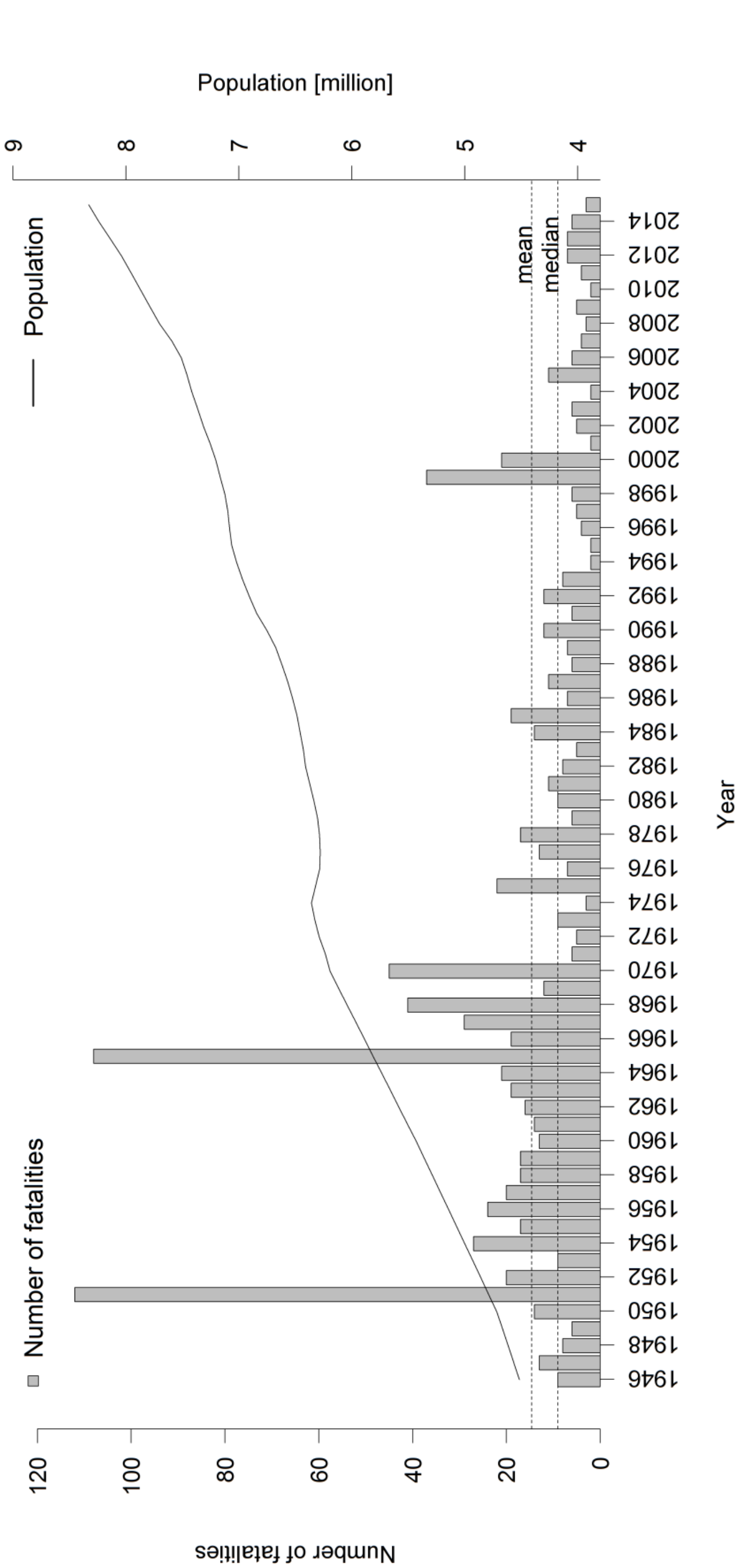

**Figure 2: Annual frequency of total natural hazard fatalities in Switzerland from 1946 to 2015. The thin black line indicates the population increase during the study period. The horizontal lines indicate the mean and median annual fatality frequencies over the entire study period.**

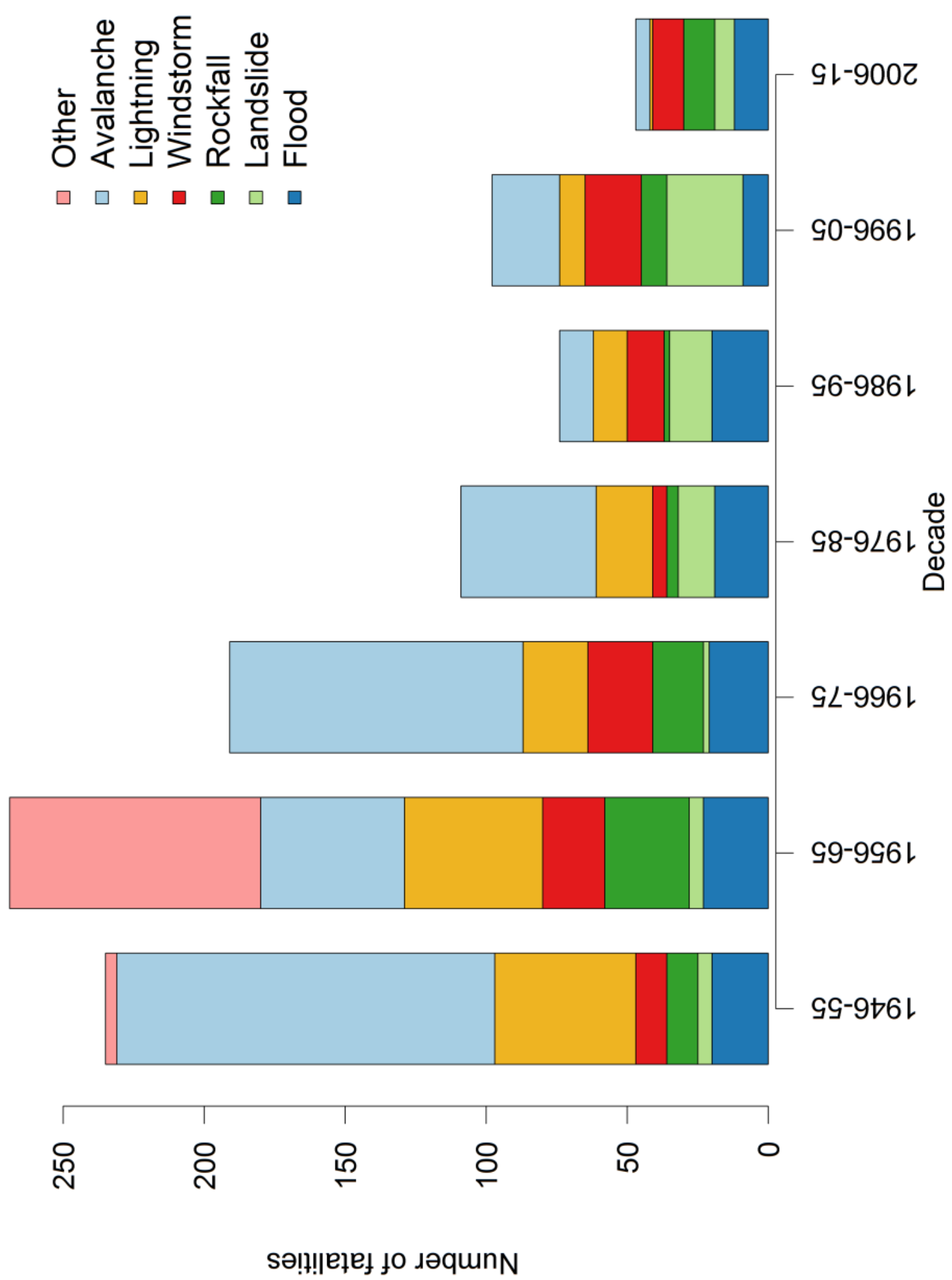

**Figure 3: Decadal sum of natural hazard fatalities in Switzerland from 1946 to 2015. The different colours indicate the seven process categories defined for this study (see section 3.4).**

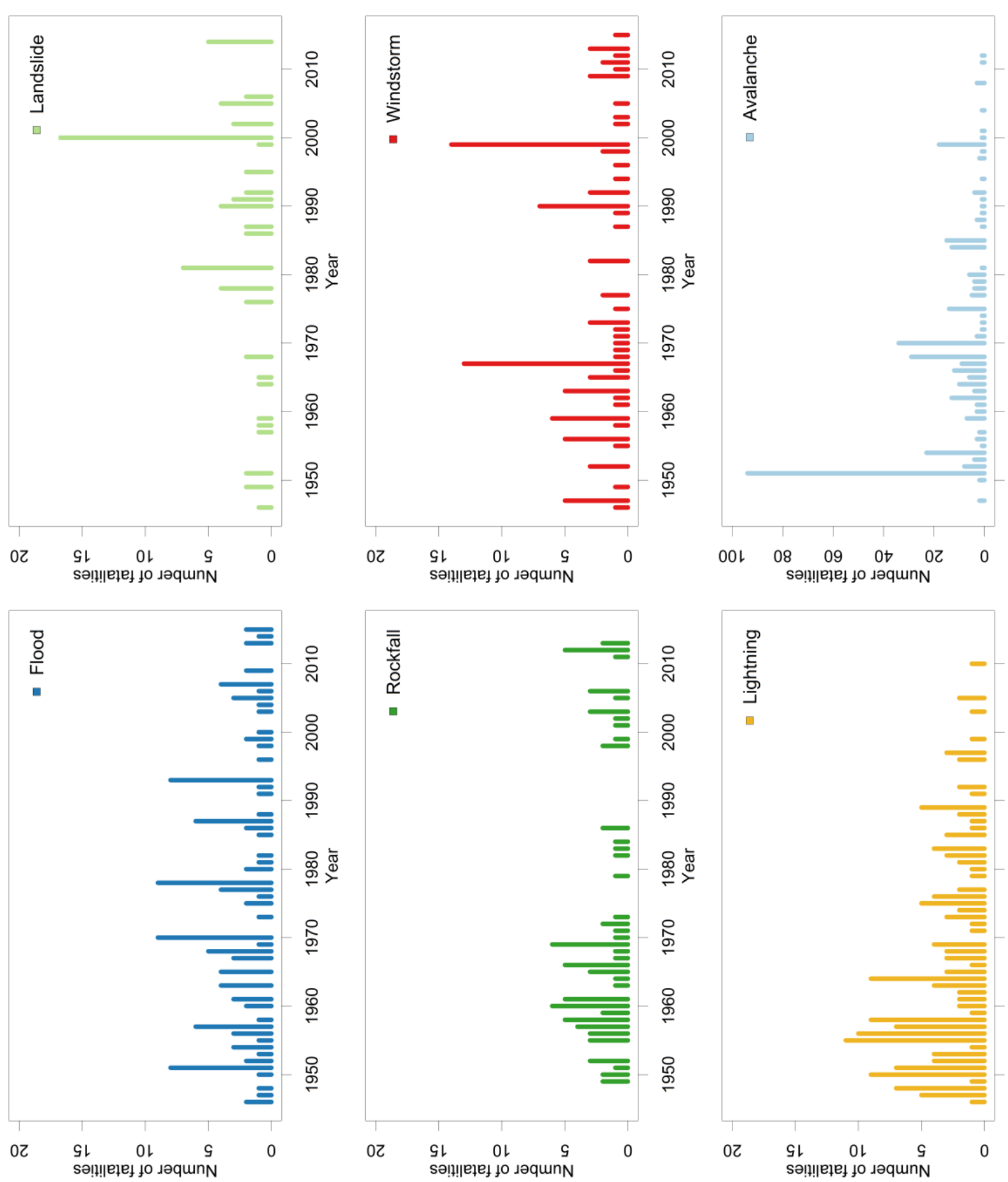

**Figure 4: Annual frequency of fatalities in Switzerland for the different natural hazard categories considered in this study (except the category other). Note that the y-axis for avalanche ranges from 0 to 100 fatalities while the y-axis for all other process categories ranges from 0 to 20 fatalities.**

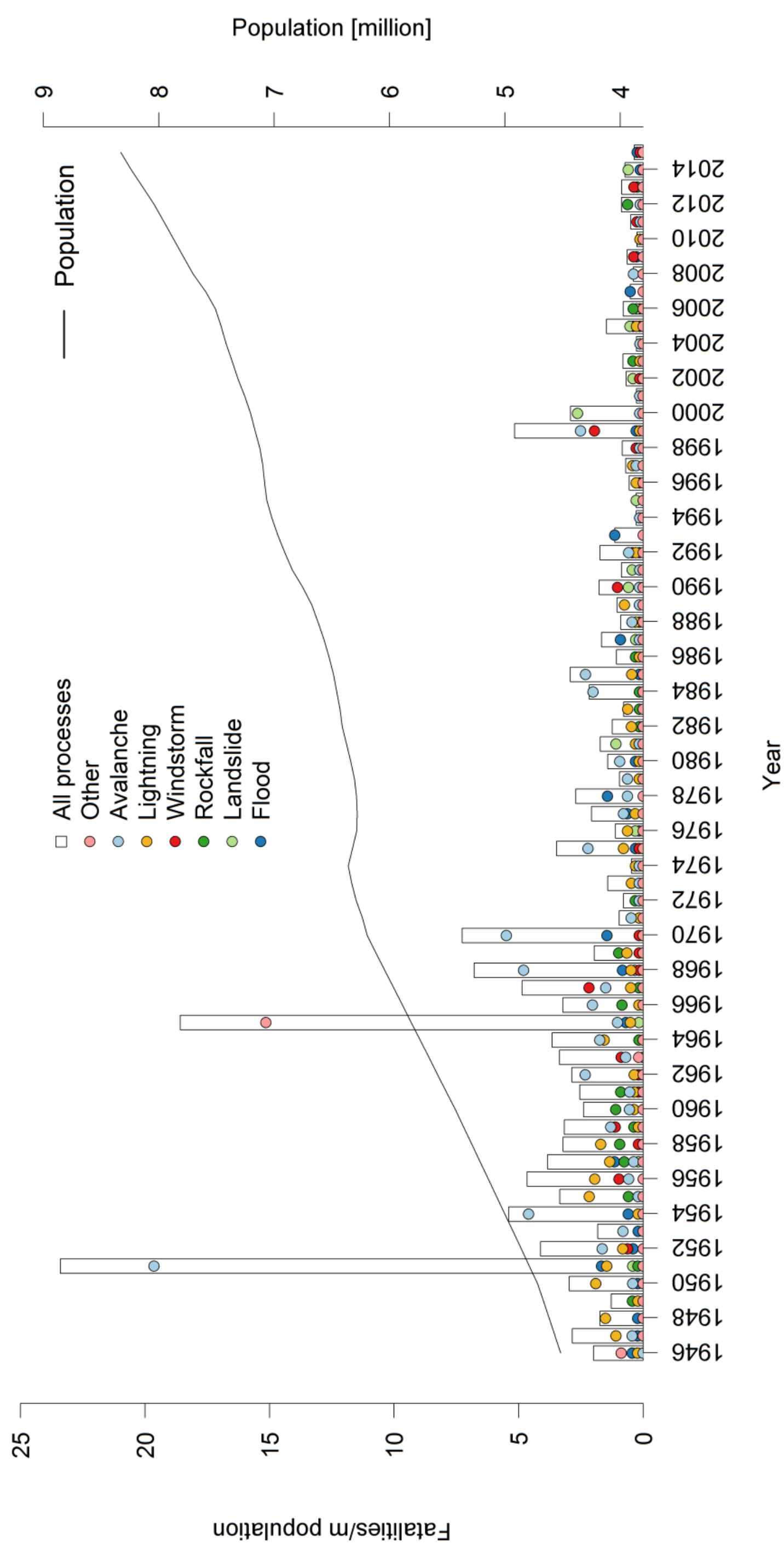

**Figure 5: Annual frequency of natural hazard fatalities in Switzerland from 1946 to 2015, normalized by population (per million population). While the transparent bars show the overall normalized fatalities, the coloured dots represent the different process categories. The thin black line indicates the population increase during the study period.**

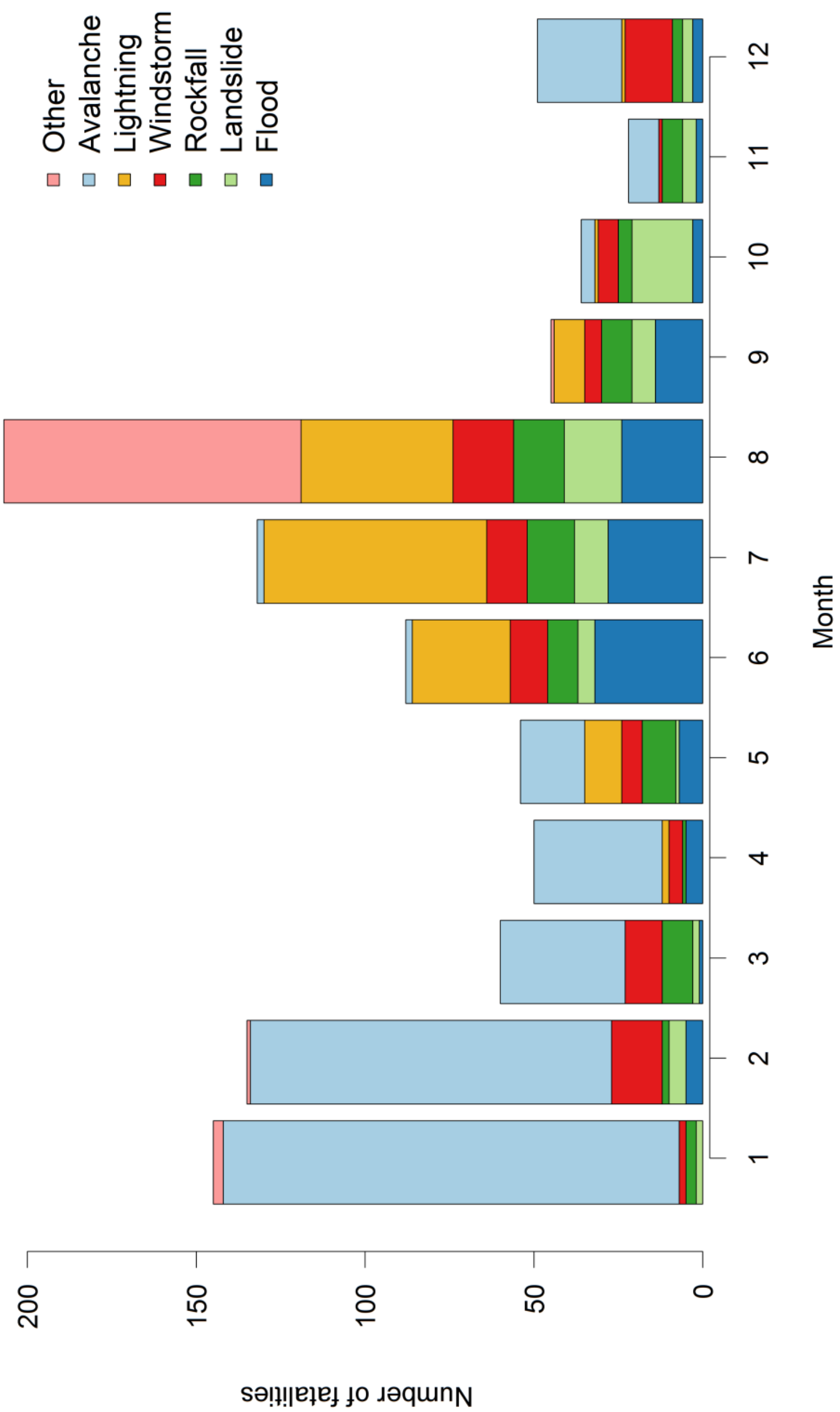

**Figure 6: Monthly distribution of natural hazard fatalities in Switzerland from 1946 to 2015 (sum over study period). The different colours indicate the seven process categories defined for this study.**

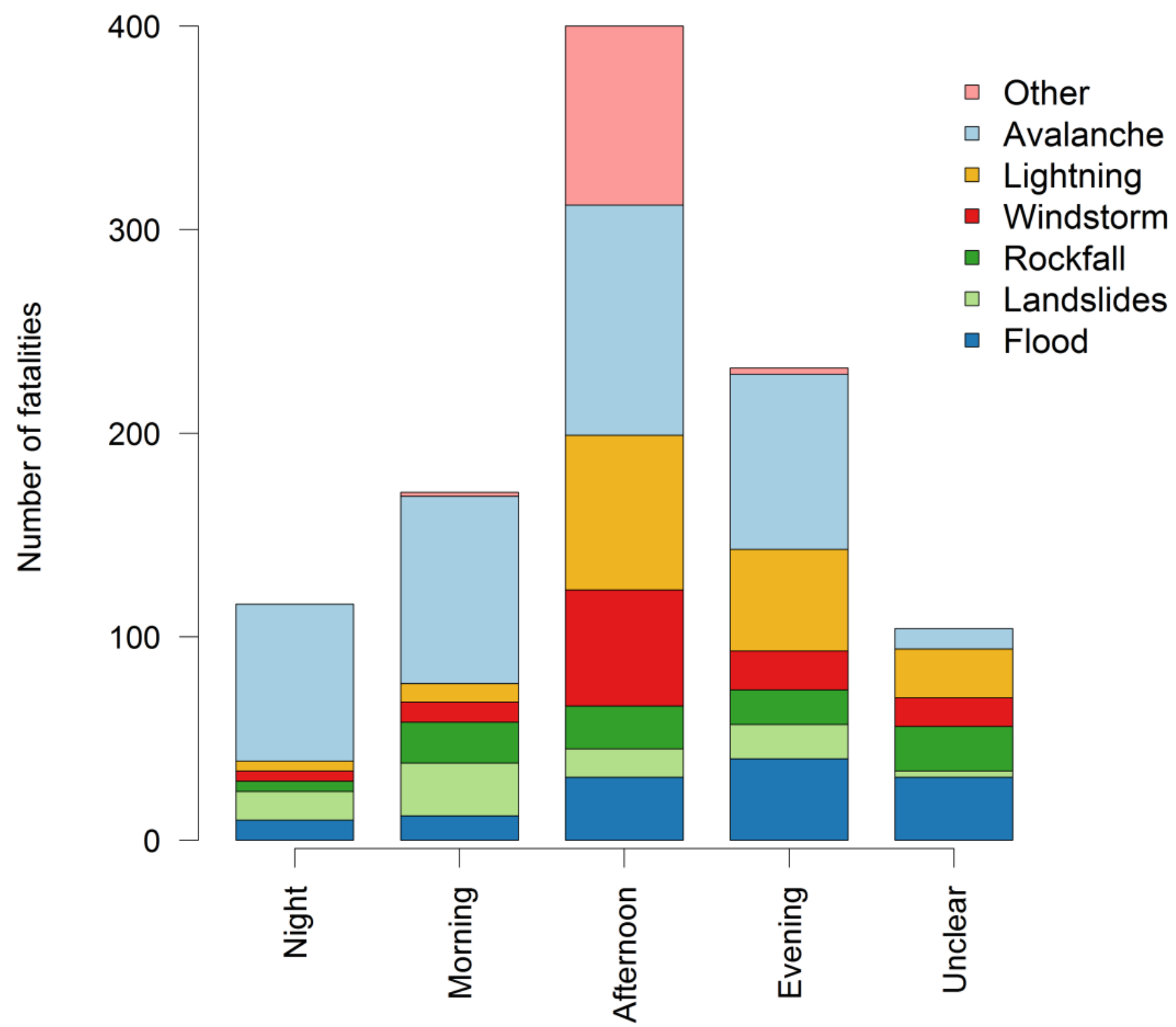

**Figure 7: Distribution of natural hazard fatalities in Switzerland from 1946 to 2015 by time of day (sum over study period); morning (06:00-11:59 local standard time), afternoon (12:00-17:59), evening (18:00-23:59) and night (00:00-05:59). The different colours indicate the seven process categories defined for this study.**

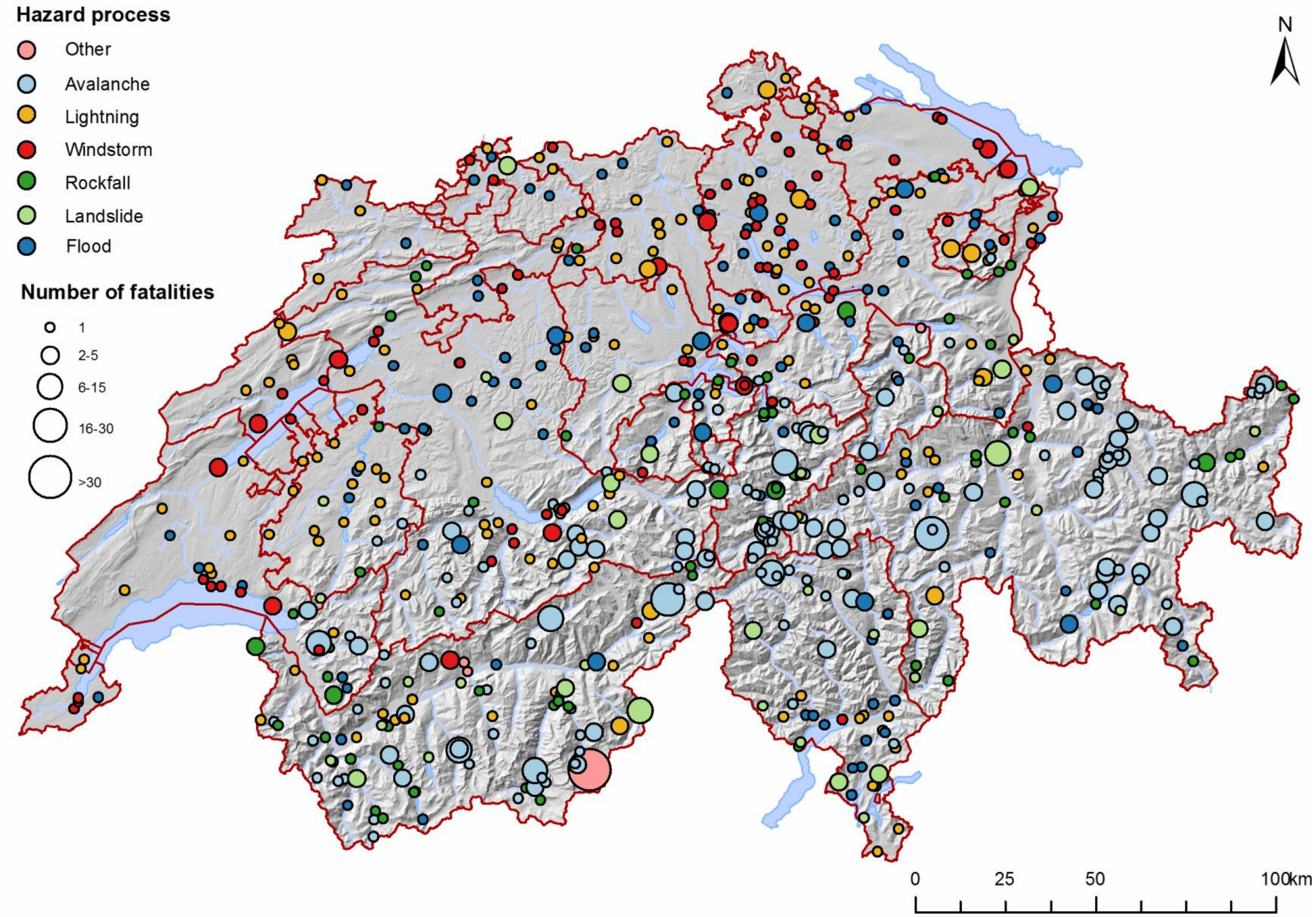

**Figure 8: (a) Spatial distribution of fatalities caused by natural hazard processes in Switzerland from 1946 to 2015. The colour of each data point indicates the process type, and the size of the symbol shows the number of deaths per fatal accident.**

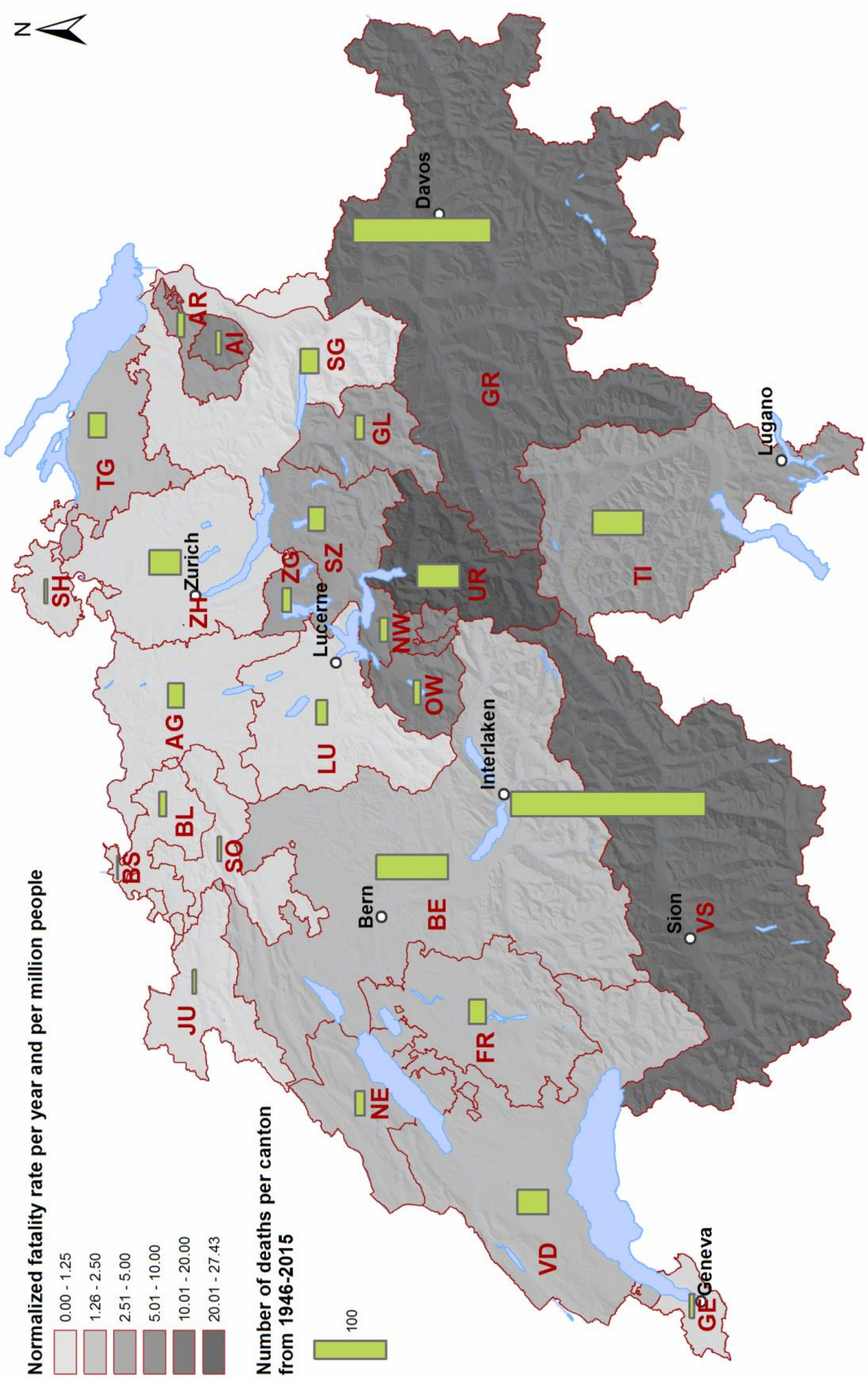

**Figure 8: (b) Spatial distribution of normalized natural hazard fatalities (per year and per million population) over the 70-year study period at the cantonal level. The bars indicate the total number of deaths for each canton over the entire period.**