# Peer review of "Natural hazard fatalities in Switzerland from 1946 to 2015"

_Natural Hazards and Earth System Sciences, 2016_

## Referee Comment (RC1) · Anonymous Referee #1 · 10 Aug 2016

**1   Summary of the article**

The study presents an explorative review on fatalities related to floods, landslides, rockfall, lightning, windstorm, snow avalanches and other geophysical hazards in Switzerland.   The data set is based on two national databases; one focusing on fatalities of floods, landslides and rockfall spanning a period from 1972 to 2015 and the other on fatalities of snow avalanches ranging from 1936 to 2015. By conducting an archival search of the Swiss national newspaper Neue Züricher Zeitung, a third database for windstorm, lightning and earthquake fatalities was generated by the authors and also used to extend the period of the first mentioned database back to the year 1946.  The combination of these three sources served as the basis for the descriptive analysis presented.

In the first part, the authors are focusing on the temporal dynamics of the fatality count. Their emphasis is on detecting trends in the total fatality count per year as well the count per hazard type per year. To quantify assumed trends, a non-parametric trend test as well as linear regression – which is in that case not the best bet – is used and confirms the trend apparent in figure 2. Beside the total count of fatalities also the crude mortality rate is shown for their study period on a yearly resolution further confirming a downward trend. Furthermore, monthly and daily frequencies are analyzed explorative. Additionally, the authors show spatial trends and comment on them in a descriptive way on the different influences of social factors such as age, location and gender.

The last part of the article is used to explain the patterns in the data and to compare the crude mortality rates to other causes of death like traffic or railroad accidents in Switzerland but also for similar hazards from other countries.

In general, the study is interesting to read, and given the scope of the target journal the contribution should be considered for final publication. The main contribution of the article is the dataset itself, another confirmation of the downward trend of fatalities due the analyzed hazard types for societies like Switzerland and the findings about the marginal importance of natural hazards as a cause of death in comparison to other causes like road accidents. However, there are some major items that caught my eyes and that should be further discussed and considered before the manuscript may become acceptable for publication.

**2   Major points of critisism**

1. The use of the whole population to calculate mortality rates is (technically and methodologically) misleading and may lead to a strong underestimation of the mortality rate. Although it is used by many authors working on that topic I think using a better suited definition of the population would greatly enhance the value of the study. To put it in a different manner: imagine a building with three floors. In the third floor, which is hermetical separated from the other two floors, a laboratory is situated in which new and highly deadly poison is tested. If there is an accident the poison would only spread in the third floor. Calculating now the mortality rate based on the all the people in the building would underestimate the real risk because only those working on the third floor are at risk.

2. The artificial political borders to agglomerate the data are a point that is directly related to this criticism, in geostatistics also well-known as MAUP. To analyze the spatial distribution of fatalities and the mortality rate, the authors should find a better solution (such as e.g. a regular grid raster).

**3   Detailed review**

In the following sections I will address in greater detail the points that lead to the aforementioned two points of major criticism refering to the sections of the stuy. When I am referring to a specific line in the article I use a `typewriter font` to detect them more easily.
**3.1  Introduction (section 1 in the article)**

In the first section an introduction to the topic and an overview on existing studies of fatalities due to different hazard types is given. Here it would be of added value if the authors could also include results from other Alpine countries, since a comparison with mortality rates of e.g. the US or some least-developed countries does not seem to be very targeted. For Switzerland, also the works of Schneebeli et al. (1997), Laternser and Schneebeli (2002) or Wilhelm (1997, see Table page 76 based on Laternser et al. 1995) should be acknowledged. For Austria, examples include those of Oberndorfer et al. (2007) or Luzian and Eller (2007), Luzian (2002) or Fuchs (2009). The section is continued by an overview on which spatial scales past studies were carried out and then looks in greater detail at studies regarding specific hazards typesjonk. At the end the authors are summarizing past work on fatalities caused by natural hazards in Switzerland and introduce the structure of their paper. For the different groups of hazards the authors also present some key findings.

**3.1.1  Flood**

The assessment of flood incidents is based on the work of Jonkman (2005) reporting an average fatality rate of 0.012 (1.2 killed of 100 affected) where the rate is fairly constant for all continents but different for different types of flooding. The most deadly ones are flash floods with a rate of 0.0362 (3.62 killed by 100 affected) followed by river floods with 0.0292 (2.92 killed by 100 affected) and drainage problems with only 0.0019 (1.9 killed by 1000 affected).

Coastal floods where excluded from the analysis by Jonkman (2005)but the authors gave a reference to Chowdhury et al. how described a tidal surge caused by a cyclon which killed 67,000 people. Here I do not think that this reference is enough to give the statement enough confidence – there many other scientific reports about deaths caused by storm surges around that should be mentioned. At the end of this section, reports on four countries about the deaths caused by flooding are presented with no further explanation.

**3.1.2 Landslides**

With respect to landslides, the authors are first referring to the study of Petley (2012) who compiled a worldwide database of non-seismically triggered landslides that resulted in a loss of life from 2004 to 2010. A total of 2620 fatal landslide resulted in 32,322 deaths which is 4.3 times greater than those recorded in the EM-DAT (7431). After the general worldwide study of Petley the authors summarize an article from Dowling and Santo from the year 2014 which is explicitly focusing on debris flows. With 213 debris flows causing 77,779 fatalities the authors yielding an average of 365 fatalities per fatal debris flow. This extraordinary high number is caused by two massive events which make up nearly 50 percent of all fatalities recorded by the two authors. The median number may therefore better reflect the number of fatalities with 11 per fatal debris flow but also this number may still be very high, compared to the European Alps. To give an example, in Fuchs and Zischg (2013) an average of around 1.5 fatalities due to torrential processes is recorded annually for the Austrian Alps. The last example of analysis of fatalities caused by landslides is given in Guzetti (2000) where the same data is used and combined with data about flooding from an article of Guzetti et al. (2005). The mortality rate for floods was estimated to be 0.053 and for landslides 0.084 for the entire study period of 1861 to 2002.
An important finding is the kind of distribution the number of fatalities may show. Petley (2012) and Guzetti et al. (2005) are both fitting a power law to the data. It is not perfectly fitting in the case of the Italian data and Petley is not making any statistical tests to quantify his visual assumptions. But there is a strong indication for the power law which in turn has implications for risk assessment.

The reference to the study about landslides in South America (Sepuleva and Petley 2005) may be deleted or more information on reported findings should be given. It would also be more convenient to refer to this study before Guzetti (2000) because of its spatial scale focusing on a continent instead on worldwide data like in Dowling and Santi (2014) and Petley (2012) but on a higher scale than Guzetti (2000) who is focusing on the national scale.

**3.1.3 Meteorological**

Focusing on meteorological hazards, the authors are reporting studies about tornados and hurricanes on a national level. Tornados are not very common in the study region of the authors. Referring to a report of the meteorological survey of Switzerland the frequency is one to five tornados per 10 years (0.1 to 0.5 per year) which is large enough to cause serious damage. In this report four tornados are described in more detail (1890, 1934, 1926, 1971) where three of them caused fatalities. With respect to a comparison to Switzerland, of more interest are the studies about non-tornadic convective and non-convective storm events carried out by Ashley and Black (2008) and Black and Ashley (2010). For the latter, more than half of the deaths are associated with aviation accidents while for the other ones most people died in vehicles or in boats which is a finding that is also reflected in the data of windstorms for this study.

Beside extreme winds the authors are also looking at the fatalities caused by lightning. They refer to studies which were carried out in different countries.

**3.1.4 Other**

Because of the unimportance of hazards such as volcanic eruptions, earthquakes and tsunamis in Switzerland the author are not further considering geophysical hazards but mention an earthquake as an example of an extreme high death toll later in the discussion.

**3.2 Data (section 3 in the article)**

The authors made a clear description of the data used, and their approach for validating the data collection before 1972 was done in a good scientific manner. Later in the discussion the authors were also talking about the drawbacks of the data collection. I would like to jump to that section (section 5.1 in the article) and express some of my concerns.

The discussion starts with six points regarding the quality of the data

1. fatality not reported

2. spatial bias because of the language of the newspaper

3. wrong selection of keywords

4. time spans with no newspaper available

5. technical problems because of bad scans

6.  wrong hazard type

The authors then conclude, based on their validation strategy, that only a small fraction of 10% of all fatal hazard events were missed. At least three of the main reasons that impair the quality of the data are subject to the first 35 years of the time span analyzed. The validation was carried out for the years 1986 to 1995. If all newspaper of that period where already digitally available problem 5 could not occur. If there were no gaps also problem 4 would be underestimated. Finally problem 6 could also be of lesser importance than in the period before 1986. So maybe expanding the validation period could damp these concerns.

3.3   Results temporal analysis (section 4.2 in the article)

First general statistics about the data are given. These mostly basic statistics are then used to describe the importance of multi-fatality events. The short sentences about the number of events per year may be enforced by adding the number of events to figure 2 (see also comments on figure 2 in section 4.2 of this review).

The determination of a trend in the data is done in two ways:  with the non-parametric Mann-Kendall test and by checking the significance of the slope of a linear regression. The choice of the non-parametric test is a good one but the linear regression performed on the normal scale of the data is not appropriate for count data and I suggest to use a Poisson regression or a regression on the natural log scale if the authors still want to use the slope as an indicator of the trend.

The fatality rate is used wrongly in `line 283` (and in other places) since according to my understanding the authors are referring to a mortality rate. Because it is not age-adjusted the term crude death rate would be in better accordance with epidemiological terms.

**3.4 Results monthly distribution (section 4.2.2 in the article)**

The key findings are the two peaked distribution of fatal natural hazards in Switzerland. The winter season dominated by avalanches and the summer season by lightning. The authors mentioned earlier that they see declining and rising trends in the annual temporal distribution and it may contribute to the monthly results if the authors would check for trends in seasonality also displaying figure 6 as a - maybe paneled - seasonal plot or adding some words to the text.

**3.5 Results spatial distribution (section 4.3 in the article)**

The authors explained in a clear manner the spatial distribution of the fatalities caused by the different hazard types. They also refer to the geomorphological region which makes their introduction at the former chapter plausible. A suggestion would be to use different maps for every hazard type and to count the number of fatalities divided by the number of events in a predefined raster or other geometrical grid. The authors may ask why my criticism is again aiming at the graphical representation this is because of the explorative nature of the topic. Paneling the hazard types would go perfectly with the excellent structured text of that chapter.

The next paragraph is dealing with the distribution of mortality rates per political region. My main concern about this result is the use of the political borders as an artificial subdivision of the space with no further connection to the underlying nature of the hazards. As an example the Canton of Bern has more than the half of its population in the five largest cities (Bern, Biel, Brugdorf, Interlaken, and Thun). Dividing the fatalities by the population incorporates a lot of people not even at risk to individual hazard types, and may therefore bias the shown results.

[Figure]

**3.6 Results regarding different social factors (section 4.4 in the article)**

The results section is nearly completed with the analysis of different factors, like the age or the accident circumstances. This section boils down to the presentation of different $n \times n$ tables which are then described by words. Using tables in transporting information is mostly not a good choice (transporting data it is one of the best). To be sure that my critic is not just the normal table-bashing I got the data of table 2 and made a circos plot which I printed out and had it a the side while reading the text. In this fashion the text transports more information and is better understood than with the table.

So I strongly recommend to find prober graphical ways (mosaic, circos tec.) to present the categorical data from table 2 and 3 (see attached figure 3 to this review).

**3.7 Disscussion fatality numbers (section 5.3 in the article)**

At `line 529` the authors state that they assume that about half of all flood deaths can be ascribed to inappropriate behavior, for example when victims are carried away by floodwaters or surprised in their home by rapidly intruding surface water in contrast at `line 585` being surprised is not considered inappropriate.

How could being surprised be an inappropriate behavior when the water is also rapidly intruding?

**3.8 Disscussion about the historical events in Switzerland**

The events of the database are by far not the most devastating that occurred in Switzerland; three events are described in detail that caused 1500, 600 and 457

lives. Alltough this is an interesting information I do not see the direct connection to the article that is primarily concerned about trends and proportions in the fatality record.

Maybe this section should be presented in a more condensed way in the introduction.

**4 Comments on figures**

I liked how the authors stayed coherent with the use of colors and I can literally feel their agony in choosing the right figures that capture all the information they want to transport. The presentation of count data is difficult as is the analysis of count data. There are few standard techniques of handling count data graphically: the bar, cumulative, scatter or line chart. The authors decided to go with the bar chart which is in my opinion not the perfect solution although this might be a question of taste.

**4.1 Figure 1 of the article**

The authors use this figure to set the spatial frame of the study introducing important categories like the cantons and the geomorphological classes of Switzerland. The hillshade was a good choice because it gives a good impression of the topography, only the use of the blue for the Swiss Plateau may be problematic because of the low distance in color space to the rivers and lakes.

**4.2 Figure 2 of the article**

The authors want to introduce their database and the trends in the number of population and the number of fatalities. I like how they plotted the mean and median

which enforces the understanding of their data and has a direct link to the text. Another important point about that plot is that the authors use it to show the declining trend in the number of fatalities as well that the first 35 years include 73% of all fatalities.

To get this information clearly transported I would suggest the following: the use of a second axis to show the population growth does not contribute to the topic – to be honest I find it distracting. By erasing it, the figure would get clearer. The most prominent feature of the figure is given by the two extremes in the year 1951 and 1965 with the high number of deaths. These two years masking the trend the authors found and I would therefore use a transformation of the count. Typically the natural logarithm is chosen because of is direct connection to the link function of the Poisson regression model. Adding a running mean with the size of 10 years - to have a connection to Figure 3 - to the transformed data will enhance the trend apparent in the data.

Another point in the text is the fact that 73% of the total fatalities occurred in the first 35 year period of the database. This fact and the declining "fatality growth" will be better shown - in my opinion - using a cumulative chart of the fatalities. This chart also has the advantage that its behavior can be used to infer if a simple Poisson process is generating the fatalities – which would be a straight line. Maybe the number of events per year as a third panel below the yearly number of fatalities would help to transport the severe multi fatality arguments of the authors.

4.3   Figure 3 of the article

The authors want to show the general decline in the number of fatalities over the years, especially the decline in the deaths caused by avalanches and lightning.

I think that the information of that figure can be also transported with an enhanced version of figure 1 and a slightly changed version of figure 4.

**4.4 Figure 4 of the article**

The authors want to describe how the number of fatalities is changing over time per hazard type or more general show the temporal behavior (also that there are no fatalities at all) and the differences between the hazard types. The x-axis is the same for all plots and therefore combining these without repeating the x-axis is recommended organizing the plot as a 3 by 3 panel. A problem may arise from the higher numbers of avalanche fatalities but this could be overcome by using a square root or logarithmic transformation of the x-axis and labeling the axis with the real number of fatalities. To better detect the temporal trend adding a running mean or other smoothing would help. Also it may be better suited to plot the cumulative number instead the number per year as a step function which would also indicate when there are no fatalities.

**4.5 Figure 5 of the article**

The key massage is the decline in crude mortality rate over the years. The same critiques as for figure 2 are true for figure 5. Because of the high spread of the rate the figure gets clumped and the distinction between the processes is hard. This is especially a problem when the figure is compared to the text from `lines 286` to `288`, since I am not able to see the distinct decline in mortality rate. Maybe transforming and paneling the data would help also in this case.

**Final remark**

I believe these shortcomings are manageable by the authors, and they may wish to revise their manuscript accordingly. To underpin my remarks, I included three Figures (Agetable, Figure 2 and Figure 4) as a supplement.

**References**

Fuchs, S.: Susceptibility versus resilience to mountain hazards in Austria – Paradigms of vulnerability revisited, Natural Hazards and Earth System Sciences, 9, 337-352, 2009.

Fuchs, S., and Zischg, A.: Vulnerabilitätslandkarte Österreich, Universität für Bodenkultur, Institut für alpine Naturgefahren, Wien152, 2013.

Guzzetti, F.: Landslide fatalities and the evaluation of landslide risk in Italy, Engineering Geology, 58, 89-107, 2000.

Guzzetti, F., Salvati, P., and Stark, C.: Historical evaluation of flood and landslide risk to the population of Italy, Environmental Management, 36, 15-36, 2005.

Jonkman, S.: Global perspective of loss of human life caused by floods, Natural Hazards, 34, 151-175, 2005.

Laternser, M., and Schneebeli, M.: Temporal trend and spatial distribution of avalanche activity during the last 50 years in Switzerland, Natural Hazards, 27, 201-230, 2002.

Luzian, R.: Die österreichische Schadenslawinen-Datenbank. Forschungsanliegen - Aufbau - erste Ergebnisse, Mitteilungen der forstlichen Bundesversuchsanstalt Wien, 51 pp., 2002.

Luzian, R., and Eller, M.: Dokumentation von Lawinenschadereignissen. Lawinenberichte der Winter von 1998/1999 bis 2003/2004, BfW Berichte, Bundesforschungs- und Ausbildungszentrum für Wald, Naturgefahren und Landschaft (BfW), Wien, 65 pp., 2007.

Oberndorfer, S., Fuchs, S., Rickenmann, D., and Andrecs, P.: Vulnerabilitätsanalyse und monetäre Schadensbewertung von Wildbachereignissen in Österreich, BfW Berichte, Bundesforschungs- und Ausbildungszentrum für Wald, Naturgefahren und Landschaft (BfW), Wien, 55 pp., 2007.

Petley, D.: Global patterns of loss of life from landslides, Geology, 40, 927-930, 2012.

Schneebeli, M., Laternser, M., and Ammann, W.: Destructive snow avalanches and climate change in the Swiss Alps, Eclogae Geologicae Helvetiae, 90, 457-461, 1997.

Wilhelm, C.: Wirtschaftlichkeit im Lawinenschutz, Mitteilungen des Eidgenössischen Instituts für Schnee- und Lawinenforschung, Eidgenössisches Institut für Schnee- und Lawinenforschung, Davos, 309 pp., 1997.
* * *
[Figure]

**Fig. 1.** Suggestion for figure 2.

Flood            Landslide

Rockfall           Windstorm

Lightning          Avalanche

Yearly number of fatalities

Year

**Fig. 2.** Suggestion for figure 4.

**Fig. 3.** Suggestion for a graphical display of the social data presented in tables.

---

## Referee Comment (RC2) · R. Holle (Referee) · 6 Sep 2016

Review of

Natural Hazard Fatalities in Switzerland from 1946 to 2015 nhess-2016-232

by Badoux, Andres, Techel, and Hegg 05 September 2016

General comments

I have reviewed the paper nhess-2016-232 "Natural Hazard Fatalities in Switzerland from 1946 to 2015" by Badoux et al. for publication in Natural Hazards and Earth System Sciences. This paper summarizes a wide variety of Swiss natural hazards since 1946, and also includes context on these phenomena around the world over many prior years. There are few such extensive, useful, and well-presented results

in this topic area. My particular interest is in the lightning area, and that topic is an emphasis of mine as noted below.

One of the general results is that younger males tend to be the most frequent victims of all types of natural disasters. It is mentioned that work scenarios are the dominant issue, but it is also stated that young males tend to be more risk-takers (line 559). It is apparent from other lightning studies that risk taking is more likely to be the dominant issue in the United States, at least.

Another comment is that many databases of natural hazards start the threshold at ten people affected per event. Instead, many phenomena, including lightning, impact one person at a time. The large number of such single-fatality incidents can exceed the total of ten-plus events. This causes an under-appreciation of several natural hazards in many reporting hazard systems. In fact, such limits affect policy as to what is being warned for the public. This is not an easy issue to resolve, but rightly is identified on line 635 in the paper.

Specific Comments

1. Confusing comments near end: Line 211 states that the lightning total "includes all people who died after being struck by lightning." Tables 1, 2, and 3 show 164 lightning fatalities. However, on page 21, the first paragraph of the Conclusions states that some lightning deaths were not included in the previous data summary. Am I reading this wrong, or has a group been excluded from the preceding results? Do the data presented earlier in the paper not include those in connection with high-risk sports and other situations (line 718)? If so, what is the real number of lightning fatalities Switzerland, or is this an extra comment that doesn't affect the earlier numbers?

2. Add global summary of fatality rates: Several lightning fatality studies are referenced starting with line 84. The following summary of national lightning fatality rates was not in the current version of the manuscript since it is quite recent: Holle, R.L., 2016: A summary of recent national-scale lightning fatality studies. Weather, Climate, and

Society, 8, 35-42.

3. Add reference to India fatality study: The manuscript on lines 660 and 665 references a study by Singh and Singh, who found an average of only 159 fatalities per year within India. There has been an additional study by Illiyas et al. who found 1,755 fatalities per year. The latter seems more likely in this very populous country. The reference is: Illiyas, F.T. K. Mohan, S.K. Mani, and A.P. Pradeepkumar, 2014: Lightning risk in India: Challenges in disaster compensation. Economic & Political Weekly, XLIX, 23-27.

4. Effect of buildings: Page 15, line 515 states that the reduction in lightning fatalities is partially due to building and structures attracting lightning. Cloud-to-ground lightning interception by large structures is relatively rare. Instead, it is recommended that the reason is due to more people spending more time inside lightning-safe structures compared with decades ago.

Technical Corrections

–Line 103: The word lightning has an extra e.

–Line 283: Figure 5 referenced here would be easier to read if a log scale were used, since most entries have small numbers.

–Line 729: The word lightning is missing the first n.

---

## Author Comment (AC1) · 7 Oct 2016

**Reply to referee #1**

We thank referee #1 for her/his detailed and insightful comments on our manuscript nhess-2016-232. Below we try to give detailed replies to the annotations and briefly outline the changes we intend to make in the manuscript.

2. Major point of criticism

*1) The use of the whole population to calculate mortality rates is (technically and methodologically) misleading and may lead to a strong underestimation of the mortality rate. Although it is used by many authors working on that topic I think using a better suited definition of the population would greatly enhance the value of the study. To put it in a different manner: imagine a building with three floors. In the third floor, which is hermetical separated from the other two floors, a laboratory is situated in which new and highly deadly poison is tested. If there is an accident the poison would only spread in the third floor. Calculating now the mortality rate based on the all the people in the building would underestimate the real risk because only those working on the third floor are at risk.*

We acknowledge this point. But we do not really see an alternative solution. The difficulty is that dependent on the process, other parts of the population are at risk. Here, maybe a regular grid raster as proposed in comment # 2 could be a solution. In this approach only a part of the population (according to the grid cell) will be used to calculate the mortality rates.

However, when using a grid raster, we see the following general shortcoming: A rough estimation of our data shows that approximately 55% of all fatalities occurred close to or in the victim's place of residence or the victim's municipality (unpublished data, since the acquisition was difficult). In 30-35% of the cases the victims were killed in large distance from their home or the victims were from a foreign country (for 10-15 % of the fatalities their origin was unknown or unclear). This shows that it is very difficult to come up with an adequate mortality rate, especially when choosing a high resolution grid (let's say with a resolution of a mean extent of a municipality).

We suggest to try the approach using a regular grid raster (see also our responses in the detailed review section). However, if the results are not meaningful, we suspect that the only solution would be to omit presenting mortality rates in this contribution.

*2. The artificial political borders to agglomerate the data are a point that is directly related to this criticism, in geostatistics also well-known as MAUP. To analyze the spatial distribution of fatalities and the mortality rate, the authors should find a better solution (such as e.g. a regular grid raster).*

We agree with the criticism of referee # 1. We considered the solution of using a regular grid. This would be (according to Swiss Statistics BFS) available for the last approximately 1-5 years. Older gridded population data is not available. We would like to use the current data, at least as a proxy for raster population 1946-2015. But we see some problems such as (i) non-linear population growth; (ii) differences in population growth of different cantons; (iii) grid resolution (only roughly 55% of all fatalities occurred close to the place of residence, see our response to point (1) above).

We suggest to use a raster with population data of the year 2015 and to clearly state this shortcoming in our MS.

**3.1 Introduction**

*Here it would be of added value if the authors could also include results from other Alpine countries, since a comparison with mortality rates of e.g. the US or some least-developed countries does not seem to be very targeted. For Switzerland, also the works of Schneebeli et al. (1997), Laternser and Schneebeli (2002) or Wilhelm (1997, see Table page 76 based on Laternser et al. 1995) should be acknowledged. For Austria, examples include those of Oberndorfer et al. (2007) or Luzian and Eller (2007), Luzian (2002) or Fuchs (2009).*

We agree with the referee and will add results from other Alpine countries as far as available. We thank the referee for indicating appropriate studies in Switzerland and Austria..

**3.1.1 Flood**

*Coastal floods where excluded from the analysis by Jonkman (2005) but the authors gave a reference to Chowdhury et al. how described a tidal surge caused by a cyclon which killed 67,000 people. Here I do not think that this reference is enough to give the statement enough confidence – there many other scientific reports about deaths caused by storm surges around that should be mentioned.*

We see the referee's point and will add references to coastal floods or storm surges (e.g. Jonkman et al., 2009; Gerritsen, 2005; Kure et al., 2016) and tsunamis (e.g. Doocy et al., 2007; Inoue et al., 2007; Ando et al., 2013).

**3.1.2 Landslides**

*[…] the authors summarize an article from Dowling and Santo from the year 2014 which is explicitly focusing on debris flows. With 213 debris flows causing 77,779 fatalities the authors yielding an average of 365 fatalities per fatal debris flow. This extraordinary high number is caused by two massive events which make up nearly 50 percent of all fatalities recorded by the two authors. The median number may therefore better reflect the number of fatalities with 11 per fatal debris flow but also this number may still be very high, compared to the European Alps. To give an example, in Fuchs and Zischg (2013) an average of around 1.5 fatalities due to torrential processes is recorded annually for the Austrian Alps.*

We will consider adding this comparison (and large difference) between fatalities in torrential environments of developing and advanced (such as e.g. Austria, cf. Fuchs and Zischg, 2013) countries.

*The reference to the study about landslides in South America (Sepuleva and Petley 2005) may be deleted or more information on reported findings should be given. It would also be more convenient to refer to this study before Guzetti (2000) because of its spatial scale […]*

We will follow the advice of referee #1 and delete the reference to the study by Sepúlveda and Petley (2005) here. This will also help to keep the introduction section reasonably short and concise.

**3.1.1 Meteorological**

*Focusing on meteorological hazards, the authors are reporting studies about tornados and hurricanes on a national level. Tornados are not very common in the study region of the authors. Referring to a report of the meteorological survey of Switzerland the frequency is one to five tornados per 10 years (0.1 to 0.5 per year)*

*which is large enough to cause serious damage. In this report four tornados are described in more detail (1890, 1934, 1926, 1971) where three of them caused fatalities.*

We thank the referee for this interesting information. The MeteoSwiss report states that of the four especially serious events two caused fatalities in Switzerland (one death in 1926 and three in 1934) and one event killed several people in France (1890). The one event that lies within our study period (1971) did only cause light injuries. We will add a sentence to the text stating that while hurricanes do not occur in Switzerland, tornadoes have occurred in the near past causing several fatalities.

**3.1.4 Other**
*Because of the unimportance of hazards such as volcanic eruptions, earthquakes and tsunamis in Switzerland the author are not further considering geophysical hazards but mention an earthquake as an example of an extreme high death toll later in the discussion.*

In connection with comment 3.8 of referee #1 (Discussion about the historical events in Switzerland), we are considering to considerably shorten our section 5.6 of the Discussion (Comparison of recent Swiss natural hazard fatality data with historic data) and to partly integrate it here, in the sixth paragraph of the Introduction.

**3.2 Data**
*[…] Later in the discussion the authors were also talking about the drawbacks of the data collection. I would like to jump to that section (section 5.1 in the article) and express some of my concerns.*
*1. fatality not reported; 2. spatial bias because of the language of the newspaper; 3. wrong selection of keywords; 4. time spans with no newspaper available; 5. technical problems because of bad scans; 6. wrong hazard type*
*The authors then conclude, based on their validation strategy, that only a small fraction of 10% of all fatal hazard events were missed. At least three of the main reasons that impair the quality of the data are subject to the first 35 years of the time span analyzed. The validation was carried out for the years 1986 to 1995. If all newspaper of that period where already digitally available problem 5 could not occur. If there were no gaps also problem 4 would be underestimated. Finally problem 6 could also be of lesser importance than in the period before 1986. So maybe expanding the validation period could damp these concerns.*

We agree that we could have chosen a different validation period (e.g. starting from 1972, which is the start of the Swiss Flood and Landslide damage database) or we could have opted for a longer validation period. Thereby trying to improve the quality of the search. However, we selected ten years of validation mainly to speed up the process and concentrate our efforts on the actual search. Also, we selected the years 1986-1995 because during this period more fatalities were recorded in the Swiss Flood and Landslide damage database than between 1972 and 1981 (27 vs. 20 victims). We stated in the manuscript that the overall data quality improved over time and is clearly better for the second 35 years of the study period (e.g. lines 466/7). This statement is based on two essential facts. (a) The main improvement that influences our data quality came in 1994 with the digital availability of all NZZ newspaper articles. (b) To complete the NZZ digital archive, newspaper issues older than 1994 were scanned and scans were processed using character recognition (problem no. 5 in section 5.1 of our manuscript). This effort was probably carried out by the Swiss Media Database in the last 10 to 20 years. Of course, older articles were more weathered at the time of scanning which resulted in qualitatively inferior scans. This, in turn, negatively affected character recognition.

Note that problem/point no. 6 in section 5.1 of the article should not affect the total number of fatalities detected in our search. For example, if a fatal channelized debris flow in the 1950ies was not identified as such, it was called something else by the newspaper (often e.g. "Schlammlawine" = mudflow or mud avalanche). However, it was still considered in our search, because we used all kinds of designations as keywords. Also note that we have not detected an increase in the occurrence of data gaps in our source of information (problem/point no. 4 in section 5.1).

In summary, we agree with referee #1 that the result of the methodological validation carried out for later years of the overall study period is probably not a good proxy for estimating the completeness of our database (or in other words the percentage of missed natural hazard deaths). By extending the validation period, this shortcoming could have been somewhat mitigated. However, we cannot expand the validation period at this point anymore. This would merely result in faintly different keywords combinations that would have to be run for all the newspaper issues again (months of work).

There are two main reasons why we think that the percentage of missed natural hazard fatalities in our overall study is not larger than 10 %. These are: (1) Approximately 37% of all fatalities in the data set presented here were caused by snow avalanches. Now, the destructive avalanche database is considered complete for fatality data and no additional search was carried out. This means that only less than two thirds of our data is subject to underestimation. (2) Grave events that cause several deaths are normally not just mentioned once in a newspaper. Often such events are described in numerous articles that span over a couple of days or even a few weeks. Some events are also mentioned in subsequent years for commemoration or retrospection. It is thus very unlikely that by reason of a data gap in the digital library, we did not register a single article on a multi-fatality event. Note that about 36% of all victims died in a multi-fatality event with at least three deaths (50% of all victims in an event with at least two deaths).

*3.3 Results temporal analysis (section 4.2 in the article)*

*First general statistics about the data are given. These mostly basic statistics are then used to describe the importance of multi-fatality events. The short sentences about the number of events per year may be enforced by adding the number of events to figure 2 (see also comments on figure 2 in section 4.2 of this review).*

This is a good point. And as outlined below in the section titled "Comments on figures", we will add the number of events per year in Figure 2 to strengthen our statement made in the first paragraph of section 4.2.1.

*The determination of a trend in the data is done in two ways: with the nonparametric Mann-Kendall test and by checking the significance of the slope of a linear regression. The choice of the non-parametric test is a good one but the linear regression performed on the normal scale of the data is not appropriate for count data and I suggest to use a Poisson regression or a regression on the natural log scale if the authors still want to use the slope as an indicator of the trend.*

We agree with the referee and will drop the linear regression performed on the normal scale of the data. Instead we will either use a Poisson regression and test the significance of the slope as an indicator or we will carry out a non-parametric Theil-Sen slope test (we still need to decide what suits the requirements better).

*The fatality rate is used wrongly in line 283 (and in other places) since according to my understanding the authors are referring to a mortality rate. Because it is not age-adjusted the term crude death rate would be in better accordance with epidemiological terms.*

In a previous version of the manuscript we used the term mortality rate instead of fatality rate (according to the remarks of the referee). During language check right before submission, we were encouraged to use fatality rate. We will discuss this with our consultant and plan to use the terms mortality rate (due to natural hazard processes) or crude mortality rate throughout the manuscript.

**3.4 Results monthly distribution (section 4.2.2 in the article)**
*The key findings are the two peaked distribution of fatal natural hazards in Switzerland. The winter season dominated by avalanches and the summer season by lightning. The authors mentioned earlier that they see declining and rising trends in the annual temporal distribution and it may contribute to the monthly results if the authors would check for trends in seasonality also displaying figure 6 as a - maybe paneled – seasonal plot or adding some words to the text.*

Natural hazards fatality numbers in winter (DJF) as well as in the months of March and April are dominated by avalanches (>62% for each of these months). The summer peak is definitely strongly influenced by light-ning fatalities, a proportion of 50% is, however, only reached for the month of July.
We like the suggestion of the referee to check for trends in seasonal fatality numbers over the study pe-riod. We will establish a few graphs and assess if they would fit in the article. Because the process types snow avalanches and lightning are the ones that show a clear decrease in the temporal distribution of fatalities, we expect that winter (DJF) and summer (JJA) fatalities have a strong decreasing trend over the study period. Spring (MAM) fatalities might also decrease due to the importance of avalanche accidents in these months. Autumn victims will probably not show a trend.

**3.5 Results spatial distribution (section 4.3 in the article)**
*The authors explained in a clear manner the spatial distribution of the fatalities caused by the different hazard types. They also refer to the geomorphological region which makes their introduction at the former chapter plausible. A suggestion would be to use different maps for every hazard type and to count the number of fatalities divided by the number of events in a predefined raster or other geometrical grid. The authors may ask why my criticism is again aiming at the graphical representation this is because of the explorative nature of the topic. Paneling the hazard types would go perfectly with the excellent structured text of that chapter.*

We will try to create a new figure with six panels (maps) for the different hazard types. In these maps we plan to show the fatality data in a raster, as proposed by the referee. We are still considering which resolution of the grids to choose (e.g. 5 km or 10 km). Additionally to panelling the hazard types, we will probably keep our current Figure 8(a) as an overview for section 4.3.

*The next paragraph is dealing with the distribution of mortality rates per political region. My main concern about this result is the use of the political borders as an artificial subdivision of the space with no further connection to the underlying nature of the hazards. As an example the Canton of Bern has more than the half of its population in the five largest cities (Bern, Biel, Brugdorf, Interlaken, and Thun). Dividing the fatalities by the population incorporates a lot of people not even at risk to individual hazard types, and may therefore bias the shown results.*

We will probably delete this paragraph in its actual form together with Figure 8b. We plan to explore the distribution of mortality rates per political region in an additional article for a Swiss journal (in German). Here, we will further discuss spatial distribution of fatalities by means of the new raster maps (if they prove successful) suggested by referee #1 in section 2 of the review (2. Major point of criticism).

**3.6 Results regarding different social factors (section 4.4 in the article)**

*[…] This section boils down to the presentation of different n × n tables which are then described by words. Using tables in transporting information is mostly not a good choice (transporting data it is one of the best). To be sure that my critic is not just the normal table-bashing I got the data of table 2 and made a circos plot which I printed out and had it a the side while reading the text. In this fashion the text transports more information and is better understood than with the table.*

*So I strongly recommend to find prober graphical ways (mosaic, circos tec.) to present the categorical data from table 2 and 3 (see attached figure 3 to this review).*

We thank the referee for suggesting this interesting way to display data. We promise to consider adequate solutions to present the information in the text (i.e. the data in Tables 2 and 3). We plan to keep Tables 2 and 3 in the supplementary material for readers that are interested in the exact numbers.

**3.7 Discussion fatality numbers (section 5.3 in the article)**

*At line 529 the authors state that they assume that about half of all flood deaths can be ascribed to inappropriate behavior, for example when victims are carried away by floodwaters or surprised in their home by rapidly intruding surface water in contrast at line 585 being surprised is not considered inappropriate.*

*How could being surprised be an inappropriate behavior when the water is also rapidly intruding?*

We understand why referee #1 raised this point and agree that the sentence at lines 529-531 should be rephrased. This misunderstanding is due to an inaccurate description of what we wanted to say.

Based on the descriptions given in the examined newspaper articles, we assume that a considerable number of fatalities caused during floods and inundations occur because of incautious behaviour. The first example pertains to people that are carried away by flood flows because they were standing too close to a channel. These victims approached swollen, dangerous rivers or streams because they wanted to cross them, to retrieve wood or other things, to save something or somebody, out of curiosity or for other unknown reasons. In many cases, circumspection would have saved these victims. The second example in our text (529-531) describes a situation that often occurs inside or around buildings. Like in the first example, regardless of a dangerous situation, victims take bad decisions and instead of taking refuge they expose themselves to a hazard. For example people try to save valuable belongings from the basement even though it is already inundated and the water level is rising quickly; or people try to drive their vehicles out of flooded underground car parks.

In contrast, the text at lines 585-586 applies to people killed by landslides. For both process types snow avalanches and landslide processes, about 50% of the fatalities occur in buildings. We assume that most of these people were aware of a critical situation (rainfall or avalanche hazard) and thought they were safe in a building. They had most probably not been asked by authorities to evacuate the building. Hence, we suppose they were surprised.

We will rephrase our sentence at lines 529-531 in order to better describe our point and will take care to avoid using the term *surprised*.

**3.8 Discussion about the historical events in Switzerland**

*The events of the database are by far not the most devastating that occurred in Switzerland; three events are described in detail that caused 1500, 600 and 457 lives. Although this is an interesting information I do not see the direct connection to the article that is primarily concerned about trends and proportions in the fatality record. Maybe this section should be presented in a more condensed way in the introduction.*

When preparing the article, the authors have discussed the importance of section 5.6 (Comparison of recent Swiss natural hazard fatality data with historic data) several times. Also, the original section was shortened before submission but not left out for the sake of completeness. We understand the referee's point of view and will delete this section from the discussion. Some of its content will briefly be presented in the introduction (sixth paragraph describing the significance of geophysical events in Switzerland).

4. Comments on figures

*I liked how the authors stayed coherent with the use of colors and I can literally feel their agony in choosing the right figures that capture all the information they want to transport. The presentation of count data is difficult as is the analysis of count data. There are few standard techniques of handling count data graphically: the bar, cumulative, scatter or line chart. The authors decided to go with the bar chart which is in my opinion not the perfect solution although this might be a question of taste.*

We thank the reviewer for the constructive suggestions on how to better handle our data graphically. We will change several of our figures according to the referee's advice (see below) and will presumably only keep two bar chart graphs (current Figs. 6 and 7 showing the monthly distribution and distribution of fatalities by time of day, respectively).

*4.1 Figure 1 of the article*
*The authors use this figure to set the spatial frame of the study introducing important categories like the cantons and the geomorphological classes of Switzerland. The hillshade was a good choice because it gives a good impression of the topography, only the use of the blue for the Swiss Plateau may be problematic because of the low distance in color space to the rivers and lakes.*

We will slightly adapt the colour used for indicating the Swiss Plateau in order to better represent the streams and lakes on the map.

*4.2 Figure 2 of the article*
*The authors want to introduce their database and the trends in the number of population and the number of fatalities. I like how they plotted the mean and median which enforces the understanding of their data and has a direct link to the text. Another important point about that plot is that the authors use it to show the declining trend in the number of fatalities as well that the first 35 years include 73% of all fatalities. To get this information clearly transported I would suggest the following: the use of a second axis to show the population growth does not contribute to the topic – to be honest I find it distracting. By erasing it, the figure would get clearer. The most prominent feature of the figure is given by the two extremes in the year 1951 and 1965 with the high number of deaths. These two years masking the trend the authors found and I would therefore use a transformation of the count. Typically the natural logarithm is chosen because of is direct connection to the link function of the Poisson regression model. Adding a running mean with the size of 10 years - to have a connection to Figure 3 - to the transformed data will enhance the trend apparent in the data.*
*Another point in the text is the fact that 73% of the total fatalities occurred in the first 35 year period of the database. This fact and the declining "fatality growth" will be better shown - in my opinion - using a cumulative chart of the fatalities. This chart also has the advantage that its behavior can be used to infer if a simple Poisson process is generating the fatalities – which would be a straight line. Maybe the number of events per year as a third panel below the yearly number of fatalities would help to transport the severe multi fatality arguments of the authors.*

The suggestions of referee #1 for the improvement of Figure 2 are all very helpful. We are working on an extended version of the suggested graph (Fig. 1 of the review). It will probably include:

→ A top panel with the cumulative number of fatalities in Switzerland over the study period (x-axis, logarithmic scale);

→ A second panel with the fatality data per year (logarithmic scale; including information on mean, median, as well as a running mean);

→ A third panel showing the number of events per year (logarithmic scale);

**4.3 Figure 3 of the article**

*The authors want to show the general decline in the number of fatalities over the years, especially the decline in the deaths caused by avalanches and lightning. I think that the information of that figure can be also transported with an enhanced version of figure 1 and a slightly changed version of figure 4.*

With strongly adapted and enhanced versions of our Figures 2 and 4, we think that the current Figure 3 will not be necessary anymore in the revised manuscript. Thus, we plan to remove it from the article.

**4.4 Figure 4 of the article**

*The authors want to describe how the number of fatalities is changing over time per hazard type or more general show the temporal behavior (also that there are no fatalities at all) and the differences between the hazard types. The x-axis is the same for all plots and therefore combining these without repeating the x-axis is recommended organizing the plot as a 3 by 3 panel. A problem may arise from the higher numbers of avalanche fatalities but this could be overcome by using a square root or logarithmic transformation of the x-axis and labeling the axis with the real number of fatalities. To better detect the temporal trend adding a running mean or other smoothing would help. Also it may be better suited to plot the cumulative number instead the number per year as a step function which would also indicate when there are no fatalities.*

We like the idea of a 3 by 2 panel graph. We plan to use the count data with a logarithmic (or square root) transformation of the y-axis and to add a running mean as suggested by the referee. A square root transformation of the axis of ordinates would allow us to show years with no fatalities.

We are also exploring the possibility to display the cumulative fatality numbers for all different process types over the study period in an additional figure.

**4.5 Figure 5 of the article**

*The key massage is the decline in crude mortality rate over the years. The same critiques as for figure 2 are true for figure 5. Because of the high spread of the rate the figure gets clumped and the distinction between the processes is hard. This is especially a problem when the figure is compared to the text from lines 286 to 288, since I am not able to see the distinct decline in mortality rate. Maybe transforming and paneling the data would help also in this case.*

This is a good point and we will try to adapt this figure to make it more comprehensible (similarly to Fig. 2). In a first effort to improve Figure 5, we noticed that with a transformed y-axis, the graphs for mortality rate and fatality count (in Fig. 2) look very much alike. Hence, maybe Figure 5 is not that important for the manuscript.

**References:**

Ando, M., Ishida, M., Hayashi, Y., Mizuki, C, Nishikawa, Y., and Tu, Y.: Interviewing insights regarding 790 the fatalities inflicted by the 2011 Great East Japan Earthquake, *Natural Hazards and Earth System Sciences*, 13, 2173–2187, doi:10.5194/nhess-13-2173-2013, 2013.

Doocy, S., Rofi, A., Moodie, C., Spring, E., Bradley, S., Burnham, G., and Robinson, C.: Tsunami mortality in Aceh Province, Indonesia, *Bulletin of the World Health Organization*, 85(2), 273-278, 2007.

Gerritsen H.: What happened in 1953? The Big Flood in the Netherlands in retrospect, *Phil. Trans. R. Soc. A*, 363, 1271-1291, doi:10.1098/rsta.2005.1568, 2005.

Inoue, S., Wijeyewickrema, A.C., Matsumoto, H., Miura, H., Gunaratna, P., Madurapperuma, M., Sekiguchi, T.: Field Survey of Tsunami Effects in Sri Lanka due to the Sumatra-Andaman Earthquake of December 26, 2004, *Pure and Applied Geophysics*, 164, 395-411, doi: 10.1007/s00024-006-0161-8, 2007.

Jonkman, S.N., Maaskant, B., Boyd, E., and Levitan, M.L.: Loss of life caused by the flooding of New Orleans after hurricane Katrina: analysis of the relationship between flood characteristics and mortality, *Risk Analysis*, 29(5), 676-698, doi:10.1111/j.1539-6924.2008.01190.x, 2009.

Kure, S., Jibiki, Y., Quimpo, M., Manalo, U.N., Ono, Y., Mano, A.: Evaluation of the Characteristics of Human Loss and Building Damage and Reasons for the Magnification of Damage Due to Typhoon Haiyan, *Coastal Engineering Journal*, 58(1), doi: 10.1142/S0578563416400088, 2016.

---

## Author Comment (AC2) · 7 Oct 2016

**Reply to referee R. Holle (referee #2)**

We thank R. Holle for his constructive comments. Below we give detailed replies and briefly outline the changes we intend to make in the manuscript.

General comments

*One of the general results is that younger males tend to be the most frequent victims of all types of natural disasters. It is mentioned that work scenarios are the dominant issue, but it is also stated that young males tend to be more risk-takers (line 559). It is apparent from other lightning studies that risk taking is more likely to be the dominant issue in the United States, at least.*

This is an interesting point. We will scan the available US studies on lightning fatalities in order to consider this point in section 5.4 by adding a sentence and appropriate references.

*Another comment is that many databases of natural hazards start the threshold at ten people affected per event. Instead, many phenomena, including lightning, impact one person at a time. The large number of such single-fatality incidents can exceed the total of ten-plus events. This causes an under-appreciation of several natural hazards in many reporting hazard systems. In fact, such limits affect policy as to what is being warned for the public. This is not an easy issue to resolve, but rightly is identified on line 635 in the paper.*

Thank you for bringing this up. We will add a sentence at the end of the first paragraph of section 5.7 (at line 637) stating that additionally to the underestimation of total fatalities, this problem also leads to under-appreciation of natural hazard processes with a high percentage of single-fatality events.

Specific comments

*1. Confusing comments near end: Line 211 states that the lightning total "includes all people who died after being struck by lightning." Tables 1, 2, and 3 show 164 lightning fatalities. However, on page 21, the first paragraph of the Conclusions states that some lightning deaths were not included in the previous data summary. Am I reading this wrong, or has a group been excluded from the preceding results? Do the data presented earlier in the paper not include those in connection with high-risk sports and other situations (line 718)? If so, what is the real number of lightning fatalities Switzerland, or is this an extra comment that doesn't affect the earlier numbers?*

As stated in Chapter 3.4 (lines 220-228) we omitted natural hazard fatalities where people willingly exposed themselves to a considerable danger, for example, we omitted loss of life due to high-risk sports (e.g. canoeing and river surfing performed deliberately during floods) and other outdoor activities in potentially dangerous environments, such as canyoning, mountaineering and rock climbing (which take place off of official hiking trails). We also excluded popular snow sport fatalities outside of ski resorts, such as freeriding and alpine touring, that have been described elsewhere.
In the case of the process type lightning, this means that we did not include mountaineers or rock climbers struck by lightning for several reasons, including the fact that it is typically very difficult to determine the cause of death in such cases. Based on recent information of the Swiss Alpine Club, approximately six climbers/mountaineers died from 2000 to 2013 after having been struck from lightning. Prior data is not available.

*2. Add global summary of fatality rates: Several lightning fatality studies are referenced starting with line 84. The following summary of national lightning fatality rates was not in the current version of the manuscript since it is quite recent: Holle, R.L., 2016: A summary of recent national-scale lightning fatality studies. Weather, Climate, and Society, 8, 35-42.*

We will add a reference to this new publication (i) in the list of US studies on lightning fatalities and (ii) at the end of the same paragraph in a new sentence emphasizing on the global summary for 23 countries on six continents.

*3. Add reference to India fatality study: The manuscript on lines 660 and 665 references a study by Singh and Singh, who found an average of only 159 fatalities per year within India. There has been an additional study by Illiyas et al. who found 1,755 fatalities per year. The latter seems more likely in this very populous country. The reference is: Illiyas, F.T. K. Mohan, S.K. Mani, and A.P. Pradeepkumar, 2014: Lightning risk in India: Challenges in disaster compensation. Economic & Political Weekly, XLIX, 23-27.*

Thank you for providing this alternative reference regarding lightning fatalities in India. Interestingly, the study of Singh and Singh (2015) was published after the contribution of Illiyas et al. (2014). However, the latter publication is not cited in Sing and Singh (2015). Also, the two studies use two different data sources. While Singh and Singh (2015) extracted information from a database on disastrous weather events of the India Meteorological Department, Illiyas et al. (2014) apply three different data sources: the Bureau of Indian Standards data set, data from the Disaster Update Bulletins of the Nat. Inst. of Disaster Management, and data from the National Crime Records Bureau.

The difference in fatality data from the two studies is very large (approximately one order of magnitude) and leaves us a bit confused. The fact that the slightly newer study by Singh and Singh (2015) was published in an indexed Journal (Meteorol. Appl. Of the Royal Meteorological Society) somehow supports it. In contrast, Illiyas et al (2014) was published in an non-indexed periodical. But then again, Illiyas et al. (2014) clearly state that lightning-associated fatalities have received little attention in India, leading to under-reporting of incidents and lower media coverage.

For us it is very difficult to decide which of the two investigations better describes the occurrence of lightning fatalities in India. However, the remark of referee R. Holle makes sense to us and the higher indication of lightning deaths by Illiyas et al. (2014) seems more likely. Especially when the data is compared to data from other, similarly developed countries (cf. e.g. Table 3 in Singh and Singh). We will thus add the reference of Illiyas et al. (2014) to our text and slightly adapt our statement in section 5.7.

*4. Effect of buildings: Page 15, line 515 states that the reduction in lightning fatalities is partially due to building and structures attracting lightning. Cloud-to-ground lightning interception by large structures is relatively rare. Instead, it is recommended that the reason is due to more people spending more time inside lightning-safe structures compared with decades ago.*

This phrase originates from Derek Elsom (2001). He used this argument to explain the decrease in lightning fatalities. With the expansion of urban areas there are more buildings and other structures which can attract lightning. This means that the probability that the lightning strikes the taller building or structure nearby is higher than that it strikes a person nearby. The comment brought up by referee #2 is also very valuable and will be included and emphasized in the manuscript.

Technical corrections

*Line 103: The word lightning has an extra e.*
Corrected.

*Line 283: Figure 5 referenced here would be easier to read if a log scale were used,*
*since most entries have small numbers.*
We agree with referee R. Holle and will adapt Figure 5 (this point was also raised by referee #1).

*Line 729: The word lightning is missing the first n.*
Corrected.

---

## Author Response (AR1)

Alexandre Badoux
Swiss Federal Research Institute WSL
Zürcherstrasse 111
Birmensdorf
Switzerland
badoux@wsl.ch

10th November 2016

**Submission of revised manuscript nhess-2016-232**

Dear Dr. S. Fuchs

Dear Sven please consider the revised version of our manuscript "Natural hazard fatalities in Switzerland from 1946 to 2015" for publication in Natural Hazards and Earth System Sciences.

We have addressed all the comments of the two referees and provide detailed responses and information on the changes made in the manuscript. If requested, we can gladly provide a version of the revised manuscript with highlighted track changes to assist in identifying all changes.

Thank you for considering our article for publication in NHESS.

Yours sincerely,

Alexandre Badoux
and co-authors

**Reply to referee #1**

We thank referee #1 for her/his detailed and insightful comments on our manuscript nhess-2016-232. Below we give detailed replies to the annotations and briefly outline the changes made in the manuscript.

2. Major point of criticism

*1) The use of the whole population to calculate mortality rates is (technically and methodologically) misleading and may lead to a strong underestimation of the mortality rate. Although it is used by many authors working on that topic I think using a better suited definition of the population would greatly enhance the value of the study. To put it in a different manner: imagine a building with three floors. In the third floor, which is hermetical separated from the other two floors, a laboratory is situated in which new and highly deadly poison is tested. If there is an accident the poison would only spread in the third floor. Calculating now the mortality rate based on the all the people in the building would underestimate the real risk because only those working on the third floor are at risk.*

We acknowledge this point. Certainly, the difficulty is that dependent on the process, other parts of the population are at risk. Here, maybe a regular grid raster as proposed in comment # 2 could be a solution. In this approach only a part of the population (according to the grid cell) will be used to calculate the mortality rates.

However, when using a grid raster, we see the following general shortcoming: A rough estimation of our data shows that approximately 55% of all fatalities occurred close to or in the victim's place of residence or the victim's municipality (unpublished data, since the acquisition was difficult). In 30-35% of the cases the victims were killed in large distance from their home or the victims were from a foreign country (for 10-15 % of the fatalities their origin was unknown or unclear). This shows that it is very difficult to come up with an adequate mortality rate, especially when choosing a high resolution grid (let's say with a resolution of a mean extent of a municipality).

We chose an approach using a regular grid raster. This choice is commented directly below in our response to the second major point of criticism of the referee and e.g. in our response to point 3.5 of this review.

*2. The artificial political borders to agglomerate the data are a point that is directly related to this criticism, in geostatistics also well-known as MAUP. To analyze the spatial distribution of fatalities and the mortality rate, the authors should find a better solution (such as e.g. a regular grid raster).*

We agree with the criticism of referee # 1. We considered the solution of using a regular grid to display the mortality rate. However, suitable population data for applying this approach    is available only since the 1990ies  (Swiss Statistics BFS). Older gridded population data is not available. Hence, we used the current data as a proxy for raster population 1946-2015. This is associated with a few problems such as (i) non-linear population growth; (ii) differences in population growth of different cantons; (iii) grid resolution (only roughly 55% of all fatalities occurred close to the place of residence, also see our response to the first major point of criticism of referee 1 above).

We produced a grid raster to show the mortality rate in Switzerland and used population data of the year 2015 (see new Figures 9 and 10). We deleted the original bottom paragraph of section 4.3 (in the Results chapter) of the manuscript that described the old Figure 8b. We replaced the text and now present a new paragraph that briefly describes the new Figure 10 (see also our response to point 3.5 of this review). Moreover, we added a detailed paragraph at the end of section 5.6 (in the Discussion chapter) to address the difficulties associated with the assessment of an adequate mortality rate.

3. Detailed review

*3.1 Introduction*
*Here it would be of added value if the authors could also include results from other Alpine countries, since a comparison with mortality rates of e.g. the US or some least-developed countries does not seem to be very targeted. For Switzerland, also the works of Schneebeli et al. (1997), Laternser and Schneebeli (2002) or Wilhelm (1997, see Table page 76 based on Laternser et al. 1995) should be acknowledged. For Austria, examples include those of Oberndorfer et al. (2007) or Luzian and Eller (2007), Luzian (2002) or Fuchs (2009).*

We agree with the referee and added references to studies carried out in Alpine countries at three locations: (i) in the second paragraph of the Introduction (studies at the regional/national scale); (ii) in a new short paragraph of the Introduction summarising previous work on avalanche fatalities; and (iii) in sections 5.3 and 5.5 of the Discussion.

*3.1.1 Flood*
*Coastal floods where excluded from the analysis by Jonkman (2005) but the authors gave a reference to Chowdhury et al. how described a tidal surge caused by a cyclon which killed 67,000 people. Here I do not think that this reference is enough to give the statement enough confidence – there many other scientific reports about deaths caused by storm surges around that should be mentioned.*

We see the referee's point. We added references to coastal floods or storm surges (e.g. Jonkman et al., 2009; Gerritsen, 2005; Kure et al., 2016) and to floods caused by tsunamis (e.g. Doocy et al., 2007; Inoue et al., 2007; Ando et al., 2013) to the introduction text.

*3.1.2 Landslides*
*[…] the authors summarize an article from Dowling and Santo from the year 2014 which is explicitly focusing on debris flows. With 213 debris flows causing 77,779 fatalities the authors yielding an average of 365 fatalities per fatal debris flow. This extraordinary high number is caused by two massive events which make up nearly 50 percent of all fatalities recorded by the two authors. The median number may therefore better reflect the number of fatalities with 11 per fatal debris flow but also this number may still be very high, compared to the European Alps. To give an example, in Fuchs and Zischg (2013) an average of around 1.5 fatalities due to torrential processes is recorded annually for the Austrian Alps.*

We added the comparison suggested by the referee between fatalities in torrential environments of developing and advanced (such as e.g. Austria, cf. Fuchs and Zischg, 2014) countries. The difference in median values is considerable.

*The reference to the study about landslides in South America (Sepuleva and Petley 2005) may be deleted or more information on reported findings should be given. It would also be more convenient to refer to this study before Guzetti (2000) because of its spatial scale […]*

We followed the advice of referee #1 and deleted the reference to the study by Sepúlveda and Petley (2005) here. This also helps to keep the introduction section reasonably short and concise.

*3.1.1 Meteorological*
*Focusing on meteorological hazards, the authors are reporting studies about tornados and hurricanes on a*

*national level. Tornados are not very common in the study region of the authors. Referring to a report of the meteorological survey of Switzerland the frequency is one to five tornados per 10 years (0.1 to 0.5 per year) which is large enough to cause serious damage. In this report four tornados are described in more detail (1890, 1934, 1926, 1971) where three of them caused fatalities.*

We thank the referee for this interesting information. The MeteoSwiss report states that of the four especially serious events two caused fatalities in Switzerland (one death in 1926 and three in 1934) and one event killed several people in France (1890). The one event that lies within our study period (1971) did only cause light injuries. We added a sentence to the text stating that tornadoes have occurred in Switzerland in the past causing several fatalities.

**3.1.4 Other**
*Because of the unimportance of hazards such as volcanic eruptions, earthquakes and tsunamis in Switzerland the author are not further considering geophysical hazards but mention an earthquake as an example of an extreme high death toll later in the discussion.*

In connection with comment 3.8 of referee #1 (discussion on the historical events in Switzerland), we decided to delete section 5.6 of the Discussion (Comparison of recent Swiss natural hazard fatality data with historic data) and to integrate it in the sixth paragraph of the Introduction in a very condensed form.

**3.2 Data**
*[…] Later in the discussion the authors were also talking about the drawbacks of the data collection. I would like to jump to that section (section 5.1 in the article) and express some of my concerns.*
*1. fatality not reported; 2. spatial bias because of the language of the newspaper; 3. wrong selection of keywords; 4. time spans with no newspaper available; 5. technical problems because of bad scans; 6. wrong hazard type*
*The authors then conclude, based on their validation strategy, that only a small fraction of 10% of all fatal hazard events were missed. At least three of the main reasons that impair the quality of the data are subject to the first 35 years of the time span analyzed. The validation was carried out for the years 1986 to 1995. If all newspaper of that period where already digitally available problem 5 could not occur. If there were no gaps also problem 4 would be underestimated. Finally problem 6 could also be of lesser importance than in the period before 1986. So maybe expanding the validation period could damp these concerns.*

We agree that we could have chosen a different validation period (e.g. starting from 1972, which is the start of the Swiss Flood and Landslide damage database) or we could have opted for a longer validation period. Thereby trying to improve the quality of the search. However, we selected ten years of validation mainly to speed up the process and concentrate our efforts on the actual search. Also, we selected the years 1986-1995 because during this period more fatalities were recorded in the Swiss Flood and Landslide damage database than between 1972 and 1981 (27 vs. 20 victims). We stated in the manuscript that the overall data quality improved over time and is clearly better for the second 35 years of the study period (e.g. lines 466/7). This statement is based on two essential facts. (a) The main improvement that influences our data quality came in 1994 with the digital availability of all NZZ newspaper articles. (b) To complete the NZZ digital archive, newspaper issues older than 1994 were scanned and scans were processed using character recognition (problem no. 5 in section 5.1 of our manuscript). This effort was probably carried out by the Swiss Media Database in the last 10 to 20 years. Of course, older articles were more weathered at the time of scanning which resulted in qualitatively inferior scans. This, in turn, negatively affected character recognition.

Note that problem/point no. 6 in section 5.1 of the article should not affect the total number of fatalities detected in our search. For example, if a fatal channelized debris flow in the 1950ies was not identified as such, it was called something else by the newspaper (often e.g. "Schlammlawine" = mudflow or mud avalanche). However, it was still considered in our search, because we used all kinds of designations as keywords. Also note that we have not detected an increase in the occurrence of data gaps in our source of information (problem/point no. 4 in section 5.1).

In summary, we agree with referee #1 that the result of the methodological validation carried out for later years of the overall study period is probably not a good proxy for estimating the completeness of our database (or in other words the percentage of missed natural hazard deaths). By extending the validation period, this shortcoming could have been somewhat mitigated. However, we cannot expand the validation period at this point anymore. This would merely result in faintly different keywords combinations that would have to be run for all the newspaper issues again (months of work).

There are two main reasons why we think that the percentage of missed natural hazard fatalities in our overall study is not larger than 10 %. These are: (1) Approximately 37% of all fatalities in the data set presented here were caused by snow avalanches. Now, the destructive avalanche database is considered complete for fatality data and no additional search was carried out. This means that only less than two thirds of our data is subject to underestimation. (2) Grave events that cause several deaths are normally not just mentioned once in a newspaper. Often such events are described in numerous articles that span over a couple of days or even a few weeks. Some events are also mentioned in subsequent years for commemoration or retrospection. It is thus very unlikely that by reason of a data gap in the digital library, we did not register a single article on a multi-fatality event (note that about 36% of all victims died in a multi-fatality event with at least three deaths, and50% of all victims in an event with at least two deaths). We included parts of this reasoning in section 5.1 of the manuscript to strengthen our statement concerning the completeness of the data set.

**3.3 Results temporal analysis (section 4.2 in the article)**

*First general statistics about the data are given. These mostly basic statistics are then used to describe the importance of multi-fatality events. The short sentences about the number of events per year may be enforced by adding the number of events to figure 2 (see also comments on figure 2 in section 4.2 of this review).*

This is a good point. And as outlined below in section 4 ("Comments on figures"), we added the number of events per year in Figure 2 (at the bottom) to strengthen our statement made in the first paragraph of section 4.2.1.

*The determination of a trend in the data is done in two ways: with the nonparametric Mann-Kendall test and by checking the significance of the slope of a linear regression. The choice of the non-parametric test is a good one but the linear regression performed on the normal scale of the data is not appropriate for count data and I suggest to use a Poisson regression or a regression on the natural log scale if the authors still want to use the slope as an indicator of the trend.*

We agree with the referee and will drop the linear regression performed on the normal scale of the data. Additionally to the Mann-Kendall test, we applied the non-parametric Theil-Sen slope test as slope estimator   According changes were made to the text in sections 4.2.1 and 4.2.2.

*The fatality rate is used wrongly in line 283 (and in other places) since according to my understanding the authors are referring to a mortality rate. Because it is not age-adjusted the term crude death rate would be in better accordance with epidemiological terms.*

We replaced the term *fatality rate* with the term *mortality rate* (or *crude mortality rate*) throughout the text (according to the remark of referee #1).

**3.4 Results monthly distribution (section 4.2.2 in the article)**

*The key findings are the two peaked distribution of fatal natural hazards in Switzerland. The winter season dominated by avalanches and the summer season by lightning. The authors mentioned earlier that they see declining and rising trends in the annual temporal distribution and it may contribute to the monthly results if the authors would check for trends in seasonality also displaying figure 6 as a - maybe paneled – seasonal plot or adding some words to the text.*

Natural hazards fatality numbers in winter (DJF) as well as in the months of March and April are dominated by avalanches (>62% for each of these months). The summer peak is definitely strongly influenced by lightning fatalities, a proportion of 50% is, however, only reached for the month of July.

We liked the suggestion of the referee to check for trends in seasonal fatality numbers over the study period. We hence established a 2 by 2 panel graph showing the annual frequency of natural hazard fatalities in Switzerland (1946-2015) for the four seasons DJF, MAM, JJA, and SON (new Fig. 6). Because the hazard process types snow avalanches and lightning are the ones that show a clear decrease in the temporal distribution of fatalities, we expected that winter (DJF) and summer (JJA) fatalities would have a significant decreasing trend over the study period. The statistical tests carried out for the seasonal data series confirmed this. Additionally, the number of victims that occurred in spring (MAM) also displays a statistically significant decreasing trend with time. It is weaker, however, compared to the trend detected for winter and summer fatalities. We added a few sentences to the second paragraph of section 4.2.2 to introduce and comment the new figure as well as the test results.

**3.5 Results spatial distribution (section 4.3 in the article)**

*The authors explained in a clear manner the spatial distribution of the fatalities caused by the different hazard types. They also refer to the geomorphological region which makes their introduction at the former chapter plausible. A suggestion would be to use different maps for every hazard type and to count the number of fatalities divided by the number of events in a predefined raster or other geometrical grid. The authors may ask why my criticism is again aiming at the graphical representation this is because of the explorative nature of the topic. Paneling the hazard types would go perfectly with the excellent structured text of that chapter.*

With all the new graphs in the manuscript (see also replies to the comments in section 4 of this review), we would actually like to refrain from adding another large and panelled figure to the main manuscript. However, we still created a new figure with six panels (maps) for the different hazard types. In these maps, we show the fatality data in a raster, as proposed by the referee (see below). The figure uses the same colours and practically the same legend as the upper map in the new Figure 9 and it nicely complements Figure 8 (original Figure 8a). This is why we decided to add it to the supplementary material in form of Figure S1.

We added references to this new Figure S1 in the second through fifth paragraphs of section 4.3.

[Figure]

Spatial distribution of natural hazard fatalities according to the hazard types flood, landslide, rockfall, windstorm, lightning and avalanche using a 10x10 km raster grid. The applied grid cell colours are in accordance with the colours in the new Figure 9 (above).

*The next paragraph is dealing with the distribution of mortality rates per political region. My main concern about this result is the use of the political borders as an artificial subdivision of the space with no further connection to the underlying nature of the hazards. As an example the Canton of Bern has more than the half of its population in the five largest cities (Bern, Biel, Brugdorf, Interlaken, and Thun). Dividing the fatalities by the population incorporates a lot of people not even at risk to individual hazard types, and may therefore bias the shown results.*

We deleted this paragraph in its original form together with Figure 8b. We formulated a new text paragraph describing our new Figures 9 and 10 (see response above in section 2 of this review, "Major point of criticism"). The distribution of mortality rates is now explored by means of the new raster maps and not using political borders.

**3.6 Results regarding different social factors (section 4.4 in the article)**

*[…] This section boils down to the presentation of different n × n tables which are then described by words. Using tables in transporting information is mostly not a good choice (transporting data it is one of the best). To be sure that my critic is not just the normal table-bashing I got the data of table 2 and made a circos plot which I printed out and had it a the side while reading the text. In this fashion the text transports more information and is better understood than with the table.*
*So I strongly recommend to find prober graphical ways (mosaic, circos tec.) to present the categorical data from table 2 and 3 (see attached figure 3 to this review).*

We thank the referee for suggesting this interesting way to display data. We considered different solutions to present the information in the manuscript (i.e. the data in the original Tables 2 and 3), amongst others mosaic and circos plots. We finally opted for mosaic plots and added two graphs to the article (new Figures 11 and 12). However, we decided to keep Table 3 in the article and Table 2 in the supplementary material of the manuscript for readers that are interested in the exact numbers.

The newly created Figure 11 presents age groups and gender for natural hazard fatalities in Switzerland subdivided according to the natural hazard type and replaces Table 2. The newly created Figure 12, instead, shows the victim's activity and gender subdivided according to the natural hazard category and complements Table 3.

We renounced to represent the victim's locality and mode of transport (also handled in the original Table 3) with additional mosaic plots because we do not want to unnecessarily extend the text. When discussing these two topics in section 4.5, we will rather reference the original Table 3 (which corresponds to the new Table 2).

**3.7 Discussion fatality numbers (section 5.3 in the article)**

*At line 529 the authors state that they assume that about half of all flood deaths can be ascribed to inappropriate behavior, for example when victims are carried away by floodwaters or surprised in their home by rapidly intruding surface water in contrast at line 585 being surprised is not considered inappropriate.*

*How could being surprised be an inappropriate behavior when the water is also rapidly intruding?*

We understand why referee #1 raised this point and agree that the sentence at lines 529-531 should be rephrased. This misunderstanding is due to an inaccurate description of what we wanted to say.

Based on the descriptions given in the examined newspaper articles, we assume that a considerable number of fatalities caused during floods and inundations occur because of incautious behaviour. The first example (in the original text) pertains to people that are carried away by flood flows because they were standing too close to a channel. These victims approached swollen, dangerous rivers or streams because they wanted to cross them, to retrieve wood or other things, to save something or somebody, out of curiosity or for other unknown reasons. In many cases, circumspection would have saved their lives. The second example in our text (529-531) describes a situation that often occurs inside or around buildings. Like in the first example, regardless of a dangerous situation, victims take bad decisions and instead of taking refuge they expose themselves to a hazard. For example people try to save valuable belongings from the basement even though it is already inundated and the water level is rising quickly; or people try to drive their vehicles out of flooded underground car parks.

In contrast, the text at lines 585-586 applies to people killed by landslides. For both process types snow avalanches and landslide processes, about 50% of the fatalities occur in buildings. We assume that most of these people were aware of a critical situation (rainfall or avalanche hazard) and thought they were safe in a building. They had most probably not been asked by authorities to evacuate the building. Hence, we suppose they did not expect an event.

We rephrased our sentence at lines 529-531 in order to better describe our point and took care to avoid using the term *surprised*. We also slightly sharpened our statement at lines 585-586.

**3.8 Discussion about the historical events in Switzerland**

*The events of the database are by far not the most devastating that occurred in Switzerland; three events are described in detail that caused 1500, 600 and 457 lives. Although this is an interesting information I do not see the direct connection to the article that is primarily concerned about trends and proportions in the fatality record. Maybe this section should be presented in a more condensed way in the introduction.*

When preparing the article, the authors have discussed the importance of section 5.6 (Comparison of recent Swiss natural hazard fatality data with historic data) several times. Also, the original text was shortened before submission but not left out for the sake of completeness. We understand the referee's point of view and deleted section 5.6 from the discussion. Some of its content is now briefly presented in the Introduction (sixth paragraph describing the significance of geophysical events in Switzerland).

**4. Comments on figures**

*I liked how the authors stayed coherent with the use of colors and I can literally feel their agony in choosing the right figures that capture all the information they want to transport. The presentation of count data is difficult as is the analysis of count data. There are few standard techniques of handling count data graphically: the bar, cumulative, scatter or line chart. The authors decided to go with the bar chart which is in my opinion not the perfect solution although this might be a question of taste.*

We thank the reviewer for the constructive suggestions on how to better handle our data graphically. We changed several of our figures according to the referee's advice (see below) and kept only two bar chart graphs (Figs. 6 and 7 in the original manuscript, showing the monthly distribution and distribution of fatalities by time of day, respectively).

**4.1 Figure 1 of the article**
*The authors use this figure to set the spatial frame of the study introducing important categories like the cantons and the geomorphological classes of Switzerland. The hillshade was a good choice because it gives a good impression of the topography, only the use of the blue for the Swiss Plateau may be problematic because of the low distance in color space to the rivers and lakes.*

We adapted the colour used for indicating the Swiss Plateau (orange instead of blue) in order to better represent the streams and lakes on the map.

**4.2 Figure 2 of the article**
*The authors want to introduce their database and the trends in the number of population and the number of fatalities. I like how they plotted the mean and median which enforces the understanding of their data and has a direct link to the text. Another important point about that plot is that the authors use it to show the declining trend in the number of fatalities as well that the first 35 years include 73% of all fatalities. To get this information clearly transported I would suggest the following: the use of a second axis to show the population growth does not contribute to the topic – to be honest I find it distracting. By erasing it, the figure would get clearer. The most prominent feature of the figure is given by the two extremes in the year 1951 and 1965 with the high number of deaths. These two years masking the trend the authors found and I would therefore use a transformation of the count. Typically the natural logarithm is chosen because of is direct connection to the link function of the Poisson regression model. Adding a running mean with the size of 10 years - to have a connection to Figure 3 - to the transformed data will enhance the trend apparent in the data.*
*Another point in the text is the fact that 73% of the total fatalities occurred in the first 35 year period of the database. This fact and the declining "fatality growth" will be better shown - in my opinion - using a cumulative chart of the fatalities. This chart also has the advantage that its behavior can be used to infer if a simple Poisson process is generating the fatalities – which would be a straight line. Maybe the number of*

*events per year as a third panel below the yearly number of fatalities would help to transport the severe multi fatality arguments of the authors.*

The suggestions of referee #1 for the improvement of Figure 2 are all very helpful. Accordingly, we completely redesigned Figure 2 which now includes:

→ A top panel with the cumulative number of fatalities in Switzerland over the study period (x-axis, logarithmic scale);

→ A second panel with the annual fatality data (logarithmic scale; including information on mean, median, as well as a 10-year running mean);

→ A third panel showing the number of events per year (logarithmic scale; also including a mean, and a median line, as well as a 10-year running mean).

We also added a reference to Figure 2 in each of the first two paragraphs of section 4.2.1 to point to the new features of the graph (cumulative data and event data).

**4.3 Figure 3 of the article**

*The authors want to show the general decline in the number of fatalities over the years, especially the decline in the deaths caused by avalanches and lightning. I think that the information of that figure can be also transported with an enhanced version of figure 1 and a slightly changed version of figure 4.*

With strongly adapted and enhanced versions of our original Figures 2 and 4 and in accordance with the view of referee #1, we decided that the original Figure 3 is not necessary anymore in the revised manuscript. Thus, we removed it from the article.

**4.4 Figure 4 of the article**

*The authors want to describe how the number of fatalities is changing over time per hazard type or more general show the temporal behavior (also that there are no fatalities at all) and the differences between the hazard types. The x-axis is the same for all plots and therefore combining these without repeating the x-axis is recommended organizing the plot as a 3 by 3 panel. A problem may arise from the higher numbers of avalanche fatalities but this could be overcome by using a square root or logarithmic transformation of the x-axis and labeling the axis with the real number of fatalities. To better detect the temporal trend adding a running mean or other smoothing would help. Also it may be better suited to plot the cumulative number instead the number per year as a step function which would also indicate when there are no fatalities.*

We like the idea of a 3 by 2 panel graph suggested by the referee. We used the count data with a logarithmic transformation of the y-axis and added a 10-year running mean. Because it is important to explicitly show years with no fatalities, we assigned data points.
Furthermore, we decided to add a figure to the manuscript displaying the cumulative fatality numbers for all different process types over the study period. This will help identifying the considerable decrease in avalanche and lightning fatalities in the second part of the study period.

**4.5 Figure 5 of the article**

*The key massage is the decline in crude mortality rate over the years. The same critiques as for figure 2 are true for figure 5. Because of the high spread of the rate the figure gets clumped and the distinction between the processes is hard. This is especially a problem when the figure is compared to the text from lines 286 to*

*288, since I am not able to see the distinct decline in mortality rate. Maybe transforming and paneling the data would help also in this case.*

We noticed that with a transformed y-axis, the graphs for mortality rate and fatality count (in Fig. 2) look very much alike. This is why we decided to remove the original Figure 5 from the study. Also, because (1) we show that the decreasing trend in the total number of fatalities over time is statistically significant and (2) because the Swiss population obviously increased in the last 70 years, it is not necessary to include a graph showing the decreasing crude mortality rate in the manuscript.

General comments

*One of the general results is that younger males tend to be the most frequent victims of all types of natural disasters. It is mentioned that work scenarios are the dominant issue, but it is also stated that young males tend to be more risk-takers (line 559). It is apparent from other lightning studies that risk taking is more likely to be the dominant issue in the United States, at least.*

This is an interesting point. We added a sentence and a reference (Jenensius, 2014) to consider this point in section 5.4 of the manuscript .

*Another comment is that many databases of natural hazards start the threshold at ten people affected per event. Instead, many phenomena, including lightning, impact one person at a time. The large number of such single-fatality incidents can exceed the total of ten-plus events. This causes an under-appreciation of several natural hazards in many reporting hazard systems. In fact, such limits affect policy as to what is being warned for the public. This is not an easy issue to resolve, but rightly is identified on line 635 in the paper.*

Thank you for bringing this up. We added a sentence at the end of the first paragraph of section 5.7 (at line 637) stating that additionally to the underestimation of total fatalities, this problem also leads to under-appreciation of natural hazard processes with a high percentage of single-fatality events.

Specific comments

*1. Confusing comments near end: Line 211 states that the lightning total "includes all people who died after being struck by lightning." Tables 1, 2, and 3 show 164 lightning fatalities. However, on page 21, the first paragraph of the Conclusions states that some lightning deaths were not included in the previous data summary. Am I reading this wrong, or has a group been excluded from the preceding results? Do the data presented earlier in the paper not include those in connection with high-risk sports and other situations (line 718)? If so, what is the real number of lightning fatalities Switzerland, or is this an extra comment that doesn't affect the earlier numbers?*

As stated in section 3.4 (lines 220-228) we omitted natural hazard fatalities where people willingly exposed themselves to a considerable danger. For example, we omitted loss of life due to high-risk sports (e.g. canoeing and river surfing performed deliberately during floods) and other outdoor activities in potentially dangerous environments, such as canyoning, mountaineering and rock climbing (which take place off of official hiking trails). We also excluded popular snow sport fatalities outside of ski resorts, such as freeriding and alpine touring, that have been described elsewhere. In the case of the process type lightning, this means that we did not include mountaineers or rock climbers struck by lightning for several reasons, including the fact that it is typically very difficult to determine the cause of death in such cases. Based on recent information of the Swiss Alpine Club, approximately six climbers/mountaineers died from 2000 to 2013 after having been struck from lightning. Prior data is not available.

At line 211, we deleted the word "all", because it might be misleading and because we did not use for the other process types (flood, landslide, rockfall, windstorm, and avalanche). We also delete the word "all" at line 215 to be consistent in all bullet points of section 3.4. We did not carry out additional changes in section 3.4 because we don't think it is necessary. Our point made in the paragraph above is quite clearly stated in lines 220-228.

*2. Add global summary of fatality rates: Several lightning fatality studies are referenced starting with line 84. The following summary of national lightning fatality rates was not in the current version of the manuscript since it is quite recent: Holle, R.L., 2016: A summary of recent national-scale lightning fatality studies. Weather, Climate, and Society, 8, 35-42.*

We added a reference to this new publication (i) in the list of US studies on lightning fatalities at lines 86-87 and (ii) at the end of the same paragraph in a new sentence emphasizing on the global summary for 23 countries on six continents.

*3. Add reference to India fatality study: The manuscript on lines 660 and 665 references a study by Singh and Singh, who found an average of only 159 fatalities per year within India. There has been an additional study by Illiyas et al. who found 1,755 fatalities per year. The latter seems more likely in this very populous country. The reference is: Illiyas, F.T. K. Mohan, S.K. Mani, and A.P. Pradeepkumar, 2014: Lightning risk in India: Challenges in disaster compensation. Economic & Political Weekly, XLIX, 23-27.*

Thank you for providing this alternative reference regarding lightning fatalities in India. Interestingly, the study of Singh and Singh (2015) was published after the contribution of Illiyas et al. (2014). However, the latter publication is not cited in Singh and Singh (2015). Also, the two studies use two different data sources. While Singh and Singh (2015) extracted information from a database on disastrous weather events of the India Meteorological Department, Illiyas et al. (2014) apply three different data sources: the Bureau of Indian Standards data set, data from the Disaster Update Bulletins of the Nat. Inst. of Disaster Management, and data from the National Crime Records Bureau.

The difference in fatality data from the two studies is very large (approximately one order of magnitude) and leaves us a bit confused. The fact that the slightly newer study by Singh and Singh (2015) was published in an indexed Journal (Meteorol. Appl. Of the Royal Meteorological Society) somehow supports it. In contrast, Illiyas et al (2014) was published in an non-indexed periodical. But then again, Illiyas et al. (2014) clearly state that lightning-associated fatalities have received little attention in India, leading to under-reporting of incidents and lower media coverage.

For us it is very difficult to decide which of the two investigations better describes the occurrence of lightning fatalities in India. However, the remark of referee R. Holle makes sense to us and the higher indication of lightning deaths by Illiyas et al. (2014) seems more likely. Especially when the data is compared to data from other, similarly developed countries (cf. e.g. Table 3 in Singh and Singh). We thus added the reference of Illiyas et al. (2014) to our text and slightly adapted our statement in the fourth paragraph of section 5.7.

*4. Effect of buildings: Page 15, line 515 states that the reduction in lightning fatalities is partially due to building and structures attracting lightning. Cloud-to-ground lightning interception by large structures is relatively rare. Instead, it is recommended that the reason is due to more people spending more time inside lightning-safe structures compared with decades ago.*

This phrase originates from Derek Elsom (2001). He used this argument to explain the decrease in lightning fatalities. With the expansion of urban areas there are more lightning-safe buildings and other structures. The comment brought up by referee #2 is very valuable and we included this point in the first paragraph of section 5.3 by slightly changing the formulation in the text.

Technical corrections

*Line 103: The word lightning has an extra e.*
Corrected.

*Line 283: Figure 5 referenced here would be easier to read if a log scale were used,*
*since most entries have small numbers.*
We agree with referee R. Holle. As a matter of fact, we now use a logarithmic scale in both Figures 2 and 4. Figure 5 was removed from the manuscript because it had only a limited informative value.

*Line 729: The word lightning is missing the first n.*
Corrected.

[revised manuscript text omitted]